# A Self-Attention Ansatz for Ab-initio Quantum Chemistry

**Ingrid von Glehn, James S. Spencer & David Pfau**
{ingridvg,jamessspencer,pfau}@deepmind.com

## Abstract

We present a novel neural network architecture using self-attention, the Wavefunction Transformer (Psiformer), which can be used as an approximation (or *Ansatz*) for solving the many-electron Schrödinger equation, the fundamental equation for quantum chemistry and material science. This equation can be solved *from first principles*, requiring no external training data. In recent years, deep neural networks like the FermiNet and PauliNet have been used to significantly improve the accuracy of these first-principle calculations, but they lack an attention-like mechanism for gating interactions between electrons. Here we show that the Psiformer can be used as a drop-in replacement for these other neural networks, often dramatically improving the accuracy of the calculations. On larger molecules especially, the ground state energy can be improved by dozens of kcal/mol, a qualitative leap over previous methods. This demonstrates that self-attention networks can learn complex quantum mechanical correlations between electrons, and are a promising route to reaching unprecedented accuracy in chemical calculations on larger systems.

## 1 Introduction

The laws of quantum mechanics describe the nature of matter at the microscopic level, and underpin the study of chemistry, condensed matter physics and material science. Although these laws have been known for nearly a century (Schrödinger, 1926), the fundamental equations are too difficult to solve analytically for all but the simplest systems. In recent years, tools from deep learning have been used to great effect to improve the quality of computational quantum physics (Carleo & Troyer, 2017). For the study of chemistry in particular, it is the quantum behavior of *electrons* that matters, which imposes certain constraints on the possible solutions. The use of deep neural networks for successfully computing the quantum behavior of molecules was introduced almost simultaneously by several groups (Pfau et al., 2020; Hermann et al., 2020; Choo et al., 2020), and has since led to a variety of extensions and improvements (Hermann et al., 2022). However, follow-up work has mostly focused on applications and iterative improvements to the neural network architectures introduced in the first set of papers.

At the same time, neural networks using self-attention layers, like the Transformer (Vaswani et al., 2017), have had a profound impact on much of machine learning. They have led to breakthroughs in natural language processing (Devlin et al., 2018), language modeling (Brown et al., 2020), image recognition (Dosovitskiy et al., 2020), and protein folding (Jumper et al., 2021). The basic self-attention layer is also permutation equivariant, a useful property for applications to chemistry, where physical quantities should be invariant to the ordering of atoms and electrons (Fuchs et al., 2020). Despite the manifest successes in other fields, no one has yet investigated whether self-attention neural networks are appropriate for approximating solutions in computational quantum mechanics.

In this work, we introduce a new self-attention neural network, the Wavefunction Transformer (Psiformer), which can be used as an approximate numerical solution (or *Ansatz*) for the fundamental equations of the quantum mechanics of electrons. We test the Psiformer on a wide variety of benchmark systems for quantum chemistry and find that it is significantly more accurate than existing neural network Ansatzes of roughly the same size. The increase in accuracy is more pronounced the larger the system is – as much as 75 times the normal standard for "chemical accuracy" – suggesting that the Psiformer is a particularly attractive approach for scaling neural network Ansatzes to larger,

more challenging systems. In what follows, we will provide an overview of the variational quantum Monte Carlo approach to computational quantum mechanics (Sec. 2), introduce the Psiformer architecture in detail (Sec. 3), present results on a wide variety of atomic and molecular benchmarks (Sec. 4) and wrap up with a discussion of future directions (Sec. 5).

## 2 BACKGROUND

### 2.1 QUANTUM MECHANICS AND CHEMISTRY

The fundamental object of study in quantum mechanics is the *wavefunction*, which represents the state of all possible classical configurations of a system. If the wavefunction is known, then all other properties of a system can be calculated from it. While there are multiple ways of representing a wavefunction, we focus on the *first quantization* approach, where the wavefunction is a map from possible particle states to a complex amplitude. The state of a single electron $\mathbf{x} \in \mathbb{R}^3 \times \{\uparrow, \downarrow\}$ can be represented by its position $\mathbf{r} \in \mathbb{R}^3$ and spin $\sigma \in \{\uparrow, \downarrow\}$. Then the wavefunction for an N-electron system is a function $\Psi : \left(\mathbb{R}^3 \times \{\uparrow, \downarrow\}\right)^N \to \mathbb{C}$. Let $\boldsymbol{x} \triangleq \mathbf{x}_1, \ldots, \mathbf{x}_N$ denote the set of all electron states. The wavefunction is constrained to have unit $\ell_2$ norm $\int d\boldsymbol{x} |\Psi|^2(\boldsymbol{x}) = 1$, and $|\Psi|^2$ can be interpreted as the probability of observing a quantum system in a given state when measured.

Not all functions are valid wavefunctions – particles must be indistinguishable, meaning $|\Psi|^2$ should be invariant to changes in ordering. Additionally, the Pauli exclusion principle states that the probability of observing any two electrons in the same state must be zero. This is enforced by requiring the wavefunction for electronic systems to be *antisymmetric*. In this paper, we will focus on how to *learn* an unnormalized approximation to $\Psi$ by representing it with a neural network.

The physical behavior of non-relativistic quantum systems is described by the Schrödinger equation. In its time-independent form, it is an eigenfunction equation $\hat{H}\Psi(\boldsymbol{x}) = E\Psi(\boldsymbol{x})$ where $\hat{H}$ is a Hermitian linear operator called the *Hamiltonian* and the scalar eigenvalue $E$ corresponds to the energy of that particular solution. In quantum chemistry, *atomic units* (a.u.) are typically used, in which the unit of distance is the Bohr radius ($a_0$), and the unit of energy is Hartree (Ha).

The physical details of a system are defined through the choice of Hamiltonian. For chemical systems, the only details which need to be specified are the locations and charges of the atomic nuclei. In quantum chemistry it is standard to approximate the nuclei as classical particles with fixed positions, known as the Born-Oppenheimer approximation, in which case the Hamiltonian becomes:

$$\hat{H} = -\frac{1}{2}\sum_i \nabla_i^2 + \sum_{i>j} \frac{1}{|\mathbf{r}_i - \mathbf{r}_j|} - \sum_{iI} \frac{Z_I}{|\mathbf{r}_i - \mathbf{R}_I|} + \sum_{I>J} \frac{Z_I Z_J}{|\mathbf{R}_I - \mathbf{R}_J|} \tag{1}$$

where $\nabla_i^2 = \sum_{j=1}^3 \frac{\partial^2}{\partial r_{ij}^2}$ is the Laplacian w.r.t. the $i$th particle and $Z_I$ and $\mathbf{R}_I$, $I \in \{1, ..., N_{\text{nuc}}\}$ are the charges and coordinates of the nuclei.

Two simplifications follow from this. First, since $\hat{H}$ is a Hermitian operator, solutions $\Psi$ must be real-valued. Thus we can restrict our attention to real-valued wavefunctions. Second, since the spins $\sigma_i$ do not appear anywhere in Eq. 1, we can fix a certain number of electrons to be spin up and the remainder to be spin down before beginning any calculation (Foulkes et al., 2001). The appropriate number for the lowest energy state can usually be guessed by heuristics such as Hund's rules.

While the time-independent Schrödinger equation defines the possible solutions of constant energy, at the energy scales relevant for most chemistry the electrons are almost always found near the lowest energy state, known as the *ground state*. Solutions with higher energy, known as *excited states*, are relevant to photochemistry, but in this paper we will restrict our attention to ground states.

For a typical small molecule, the total energy of a system is on the order of hundreds to thousands of Hartrees. However the relevant energy scale for chemical bonds is typically *much* smaller – on the order of 1 kilocalorie per mole (kcal/mol), or $\sim 1.6$ mHa – less than one part in *one hundred thousand* of the total energy. Calculations within 1 kcal/mol of the ground truth are generally considered "chemically accurate". Mean-field methods are typically within about 0.5% of the true total energy. The difference between the mean-field energy and true energy is known as the *correlation energy*, and chemical accuracy is usually less than 1% of this correlation energy. For example, the binding energy of the benzene dimer (investigated in Section 4.5) is only $\sim 4$ mHa.

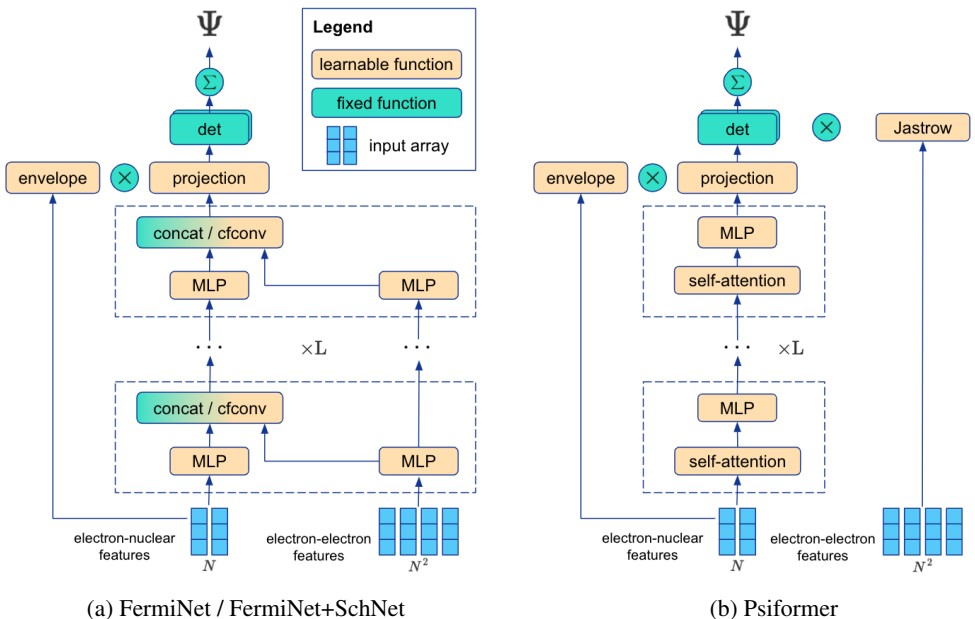

(a) FermiNet / FermiNet+SchNet  (b) Psiformer

Figure 1: Comparison of (a) FermiNet and FermiNet+SchNet and (b) the Psiformer. The FermiNet variants have two streams, acting on electron-nuclear and electron-electron features, which are merged via concatenation or continuous-filter convolution operations. In contrast, the Psiformer uses a single stream of self-attention layers, acting on electron-nuclear features only. Electron-electron features appear only via the Jastrow factor. The FermiNet+SchNet also includes a nuclear embedding stream and separate spin-dependant electron-electron streams, not pictured here.

## 2.2 VARIATIONAL QUANTUM MONTE CARLO

There are a wide variety of computational techniques to find the ground state solution of the Hamiltonian in Eq 1. We are particularly interested in solving these equations *from first principles* (*ab-initio*), that is, *without* any data other than the atomic positions. The ab-initio method most compatible with the modern deep learning paradigm is *variational quantum Monte Carlo* (VMC, Foulkes et al. (2001)). In VMC, a parametric wavefunction approximation (or Ansatz) is optimized using samples from the Ansatz itself, in much the same way that deep neural networks are optimized by gradient descent on stochastic minibatches. VMC is variational in the sense that it minimizes an upper bound on the energy of a system. Therefore if two VMC solutions give different energies, the one with the lower energy will be closer to the true energy, even if the true energy is not known.

In VMC, we start with an *unnormalized* wavefunction Ansatz $\Psi_\theta : \left(\mathbb{R}^3 \times \{\uparrow, \downarrow\}\right)^N \to \mathbb{R}$ with parameters $\theta$. The expected energy of the system is given by the Rayleigh quotient:

$$\mathcal{L}_\theta = \frac{\langle \Psi_\theta \hat{H} \Psi_\theta \rangle}{\langle \Psi_\theta^2 \rangle} = \frac{\int d\boldsymbol{x} \Psi_\theta(\boldsymbol{x}) \hat{H} \Psi_\theta(\boldsymbol{x})}{\int d\boldsymbol{x} \Psi_\theta^2(\boldsymbol{x})} = \mathbb{E}_{\boldsymbol{x} \sim \Psi_\theta^2} \left[ \Psi_\theta^{-1}(\boldsymbol{x}) \hat{H} \Psi_\theta(\boldsymbol{x}) \right] = \mathbb{E}_{\boldsymbol{x} \sim \Psi_\theta^2} \left[ E_L(\boldsymbol{x}) \right] \quad (2)$$

where we have rewritten the Rayleigh quotient as an expectation over a random variable proportional to $\Psi_\theta^2$ on the right hand side. The term $E_L(\boldsymbol{x}) = \Psi^{-1}(\boldsymbol{x}) \hat{H} \Psi(\boldsymbol{x})$ is known as the *local* energy. Details on how to compute the local energy and unbiased estimates of the gradient of the average energy are given in Sec. A.1 in the appendix.

In VMC, samples from the distribution proportional to $\Psi^2$ are generated by Monte Carlo methods, and unbiased estimates of the gradient are used to optimize the Ansatz, either by standard stochastic gradient methods, or more advanced methods (Umrigar et al., 2007; Sorella, 1998). Notably, the samples $\boldsymbol{x} \sim \Psi_\theta^2$ can be generated *from the Ansatz itself*, rather than requiring external data.

The form of $\Psi$ must be restricted to antisymmetric functions to avoid collapsing onto non-physical solutions. This is most commonly done by taking the determinant of a matrix of single-electron

functions $\Psi(\boldsymbol{x}) = \det[\boldsymbol{\Phi}(\boldsymbol{x})]$, where $\boldsymbol{\Phi}(\boldsymbol{x})$ denotes the matrix with elements $\phi_i(\mathbf{x}_j)$, since the determinant is antisymmetric under exchange of rows or columns. This is known as a *Slater determinant*, and the minimum-energy wavefunction of this form gives the mean-field solution to the Schrödinger equation. While $\phi_i$ is a function of one electron in a Slater determinant, any permutation-equivariant function of *all* electrons can be used as input to $\boldsymbol{\Phi}$ and $\Psi$ will still be antisymmetric.

The potential energy becomes infinite when particles overlap, which places strict constraints on the form of the wavefunction at these points, known as the *Kato cusp conditions* (Kato, 1957). The cusp conditions state that the wavefunction must be non-differentiable at these points, and give exact values for the average derivatives at the cusps. This can be built into an Ansatz by multiplying by a *Jastrow factor* which satisfies these conditions analytically (Drummond et al., 2004).

### 2.3 RELATED WORK

Machine learning has found numerous applications to computational chemistry in recent years, but has mostly focused on problems at the level of classical physics (Schütt et al., 2018; Fuchs et al., 2020; Batzner et al., 2022; Segler et al., 2018; Gómez-Bombarelli et al., 2018), which all rely on learning from large datasets of experiments or *ab-initio* calculations. There is also work on machine learning for density functional theory (DFT) (Nagai et al., 2020; Kirkpatrick et al., 2021), which is an intermediate between classical and all-electron quantum chemistry, but even this is still primarily a supervised learning problem and relies on *ab-initio* calculations for data. Here instead, we are focused on improving the *ab-initio* methods themselves.

For a thorough introduction to *ab-initio* quantum chemistry, we recommend Helgaker et al. (2014) and Szabo & Ostlund (2012). Within this field, VMC was considered a simple but low-accuracy method, failing to match the performance of sophisticated methods (Motta et al., 2020), but often used as a starting point for *diffusion* Monte Carlo (DMC) calculations, which are more accurate, but do not produce an explicit functional form for the wavefunction (Foulkes et al., 2001). These VMC calculations typically used a *Slater-Jastrow* Ansatz (Kwon et al., 1993), which consists of a large linear combination of Slater determinants multiplied by a Jastrow factor. They also sometimes include *backflow*, a coordinate transformation that accounts for electron correlations with a fixed functional form (Feynman & Cohen, 1956).

Recently, the use of neural network Ansatzes in VMC was shown to greatly improve the accuracy of many-electron calculations, often making them competitive with, or in some circumstances superior to, methods like DMC or coupled cluster (Pfau et al., 2020; Hermann et al., 2020; Choo et al., 2020; Han et al., 2019; Luo & Clark, 2019; Taddei et al., 2015). In first quantization, these Ansatzes used a sum of a small number of determinants to construct antisymmetric functions, but used very general permutation-equivariant functions of *all* electrons as inputs, rather than the single-electron functions used in Slater determinants. Some, like the PauliNet (Hermann et al., 2020), used Jastrow factors, while the FermiNet (Pfau et al., 2020) used non-differentiable input features to learn the cusp conditions.

Most follow-up work, as surveyed in Hermann et al. (2022), integrated these Ansatzes with other methods and applications, but did not significantly alter the architecture of the neural networks. The most significant departure in terms of the neural network architecture was Gerard et al. (2022), which extended the FermiNet – generally recognized to be the most accurate neural network Ansatz up to that point – and integrated it with several details of the PauliNet, especially the continuous-filter convolutions also used by the SchNet (Schütt et al., 2018), claiming to reach even higher accuracy on several challenging systems. We refer to this architecure as the FermiNet+SchNet.

## 3 THE PSIFORMER

The Psiformer has the basic form:

$$\Psi_\theta(\boldsymbol{x}) = \exp\left(\mathcal{J}_\theta(\boldsymbol{x})\right) \sum_{k=1}^{N_{\mathrm{det}}} \det[\boldsymbol{\Phi}_\theta^k(\boldsymbol{x})], \tag{3}$$

where $\mathcal{J}_\theta : (\mathbb{R}^3 \times \{\uparrow, \downarrow\})^N \to \mathbb{R}$ and $\boldsymbol{\Phi}_\theta^k : (\mathbb{R}^3 \times \{\uparrow, \downarrow\})^N \to \mathbb{R}^{N \times N}$ are functions with learnable parameters $\theta$. This is similar to the Slater-Jastrow Ansatz, FermiNet and PauliNet, with the key

difference being that in the Psiformer, $\boldsymbol{\Phi}_\theta^k$ consists of a sequence of multiheaded self-attention layers (Vaswani et al., 2017). The key motivation for this is that the electron-electron dependence in the Hamiltonian introduces subtle and complex dependence in the wavefunction. Self-attention is one way of introducing this without a fixed functional form. The high-level structure of the Psiformer is shown in Fig. 1(b), where it is contrasted with the FermiNet (and SchNet extension) in Fig. 1(a).

Because a self-attention layer takes a sequence of vectors as input, only features of single electrons are used as input to $\boldsymbol{\Phi}_\theta^k$. The input feature vector $\mathbf{f}_i^0$ for electron $i$ is similar to the one-electron stream of the FermiNet, which uses a concatenation of electron-nuclear differences $\mathbf{r}_i - \mathbf{R}_I$ and distances $|\mathbf{r}_i - \mathbf{R}_I|$, $I = 1, \ldots, N_{\text{nuc}}$, with two key differences. First, we found that for systems with widely separated atoms, using FermiNet one-electron features caused self-attention Ansatzes to become unstable, so we rescale the inputs in the Psiformer by a factor of $\log(1 + |\mathbf{r_i} - \mathbf{R_I}|)/|\mathbf{r_i} - \mathbf{R_I}|$, so that the input vectors grow logarithmically with distance from the nucleus. Second, we concatenate the spin $\sigma_i$ into the input feature vector itself (mapping $\uparrow$ to 1 and $\downarrow$ to -1). While this spin term is kept fixed dur-

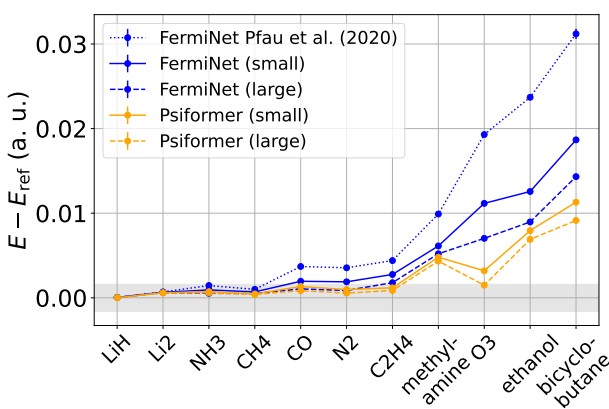

Figure 2: FermiNet and Psiformer accuracy on small molecules. Geometries and CCSD(T)/CBS reference energies are taken from Pfau et al. (2020). The grey region indicates chemical accuracy (1 kcal/mol or 1.6 mHa) relative to the reference energies.

ing training, it breaks the symmetry between spin up and spin down electrons, so that columns of $\boldsymbol{\Phi}_\theta^k$ are only equivariant under exchange of same-spin electrons. This is a notable departure from the FermiNet and PauliNet, where the difference between spin-up and spin-down electrons is instead built into the architecture.

The input features $\mathbf{f}_i^0$ are next projected into the same dimension as the attention inputs by a linear mapping $\mathbf{h}_i^0 = \mathbf{W}^0 \mathbf{f}_i^0$, and then passed into a sequence of multiheaded self-attention layers followed by linear-nonlinear layers, both with residual connections:

$$\mathbf{f}_i^{\ell+1} = \mathbf{h}_i^\ell + \mathbf{W}_o^\ell \text{concat}_h \left[ \text{SELFATTN}_i(\mathbf{h}_1^\ell, \ldots, \mathbf{h}_N^\ell; \mathbf{W}_q^{\ell h}, \mathbf{W}_k^{\ell h}, \mathbf{W}_v^{\ell h}) \right] \qquad (4)$$

$$\mathbf{h}_i^{\ell+1} = \mathbf{f}_i^{\ell+1} + \tanh\left(\mathbf{W}^{\ell+1}\mathbf{f}_i^{\ell+1} + \mathbf{b}^{\ell+1}\right) \qquad (5)$$

where $h$ indexes the different attention heads, $\text{concat}_h$ denotes concatenation of the output from different attention heads, and SELFATTN denotes standard self-attention:

$$\text{SELFATTN}_i(\mathbf{h}_1, \ldots, \mathbf{h}_N; \mathbf{W}_q, \mathbf{W}_k, \mathbf{W}_v) = \frac{1}{\sqrt{d}} \sum_j \sigma_j \left(\mathbf{q}_1^T \mathbf{k}_i, \ldots, \mathbf{q}_N^T \mathbf{k}_i\right) \mathbf{v}_j \qquad (6)$$

$$\mathbf{k}_i = \mathbf{W}_k \mathbf{h}_i, \ \mathbf{q}_i = \mathbf{W}_q \mathbf{h}_i, \ \mathbf{v}_i = \mathbf{W}_v \mathbf{h}_i \qquad (7)$$

$$\sigma_i(x_1, \ldots, x_N) = \frac{\exp(x_i)}{\sum_j \exp(x_j)} \qquad (8)$$

where $d$ is the output dimension of the key and query weights. In principle, multiple linear-nonlinear layers could be used in-between self-attention layers, but we found that adding a deeper MLP between self-attention layers was less effective than adding more self-attention layers. While a smooth nonlinearity must be used to guarantee that a wavefunction is smooth everywhere except the cusps, we found that using activation functions other than tanh had a marginal impact on performance.

A final linear projection into a $NN_{\text{det}}$ dimensional space is applied to the activations, and the output is multiplied by a weighted sum of exponentially-decaying envelopes, $\Phi_{ij}^k = \Omega_{ij}^k \mathbf{w}_i^{kT} \mathbf{h}_j^L$, where

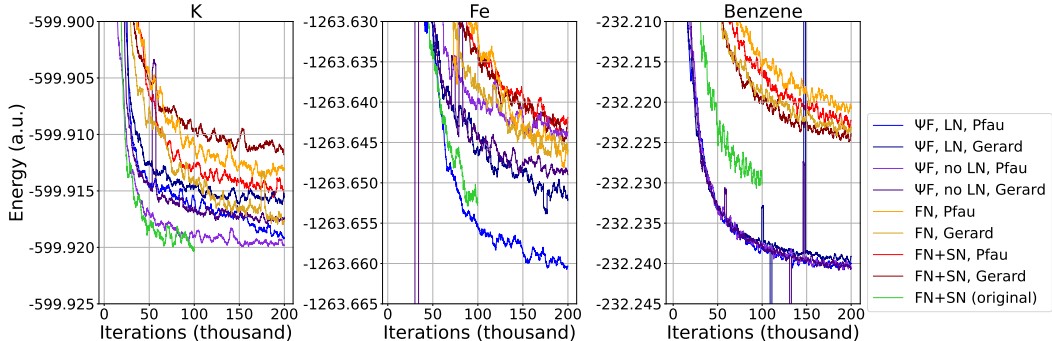

Figure 3: Comparison of the Psiformer ($\Psi$F), with and without LayerNorm (LN) against the FermiNet (FN) and FermiNet+SchNet (FN+SN) using training hyperparameters from Pfau et al. (2020) and Gerard et al. (2022). Learning curves from the original FN+SN implementation in Gerard et al. (2022) are in green. Energies are smoothed over 4000 iterations.

$\Omega_{ij}^k = \sum_I \pi_{iI}^k \exp\left(-\sigma_{iI}^k |\mathbf{r}_j - \mathbf{R}_I|\right)$. This is so that the boundary condition $\lim_{|\mathbf{r}|\to\infty} \Psi_\theta(\mathbf{r}) = 0$ is enforced, and is the same form as the envelope used by the FermiNet in Spencer et al. (2020a). The matrix elements $\Phi_{ij}^k$ are then passed into $k$ determinants and the output summed together.

As the distances $|\mathbf{r}_i - \mathbf{R}_I|$ are inputs to the Psiformer, it is capable of learning the electron-nuclear cusp conditions, much like the FermiNet. However, the self-attention part of the Psiformer does not take pairwise electron distances as inputs, so it cannot learn the electron-electron cusp conditions. Instead, the Psiformer uses a conventional Jastrow factor *only* for the electron-electron cusps. We use a particularly simple Jastrow factor:

$$\mathcal{J}_\theta(\boldsymbol{x}) = \sum_{i<j;\sigma_i=\sigma_j} -\frac{1}{4}\frac{\alpha_{\mathrm{par}}^2}{\alpha_{\mathrm{par}} + |\mathbf{r}_i - \mathbf{r}_j|} + \sum_{i,j;\sigma_i\neq\sigma_j} -\frac{1}{2}\frac{\alpha_{\mathrm{anti}}^2}{\alpha_{\mathrm{anti}} + |\mathbf{r}_i - \mathbf{r}_j|} \tag{9}$$

which has only two free parameters, $\alpha_{\mathrm{par}}$ and $\alpha_{\mathrm{anti}}$, and works well in practice.

## 4 EXPERIMENTS

Here we present an evaluation of the Psiformer on a wide variety of benchmark systems. Where it is not specified, all results with the FermiNet and FermiNet+SchNet are with our own implementation, forked from Spencer et al. (2020b). We use Kronecker-factored Approximate Curvature (KFAC) (Martens & Grosse, 2015) to optimize the Psiformer. The use of KFAC for self-attention has been investigated in Zhang et al. (2019). We show the advantage of KFAC, and how it interacts with LayerNorm, in Sec. A.2.3 of the appendix. We use hyperparameters and a Metropolis-Hastings MCMC algorithm similar to the original FermiNet paper, though we have made several modifications which help for larger systems: we pretrain for longer, we generate samples for pretraining from the target wavefunction, we take more MCMC steps between parameter updates, we propose updates for subsets of electrons rather than all electrons simultaneously, and we slightly change the gradient computation to be more robust to outliers. Details are given in Sec. A.2.4 of the appendix.

### 4.1 REVISITING SMALL MOLECULES

In Pfau et al. (2020), they compared the FermiNet against CCSD(T) extrapolated to the complete basis set (CBS) limit on a number of small molecules (4-30 electrons) from the G3 database (Curtiss et al., 2000). While the FermiNet captured more than 99% of the correlation energy relative to CCSD(T)/CBS for systems as large as ethene (16 electrons), the quality of the FermiNet calculation began to decline as the system size grew. While some of this discrepancy was reduced simply by changing to a framework with better numerics (Spencer et al., 2020a), the reported difference in energy for bicyclobutane was still greater than 20 mHa. Here we revisit these systems, comparing the FermiNet with training improvements against the Psiformer.

| System | FermiNet | | FermiNet+SchNet | | Psiformer | | Expt. |
|---|---|---|---|---|---|---|---|
| | Small | Large | Small | Large | Small | Large | |
| K (Ha) | -599.9133(2) | -599.9149(2) | -599.9153(2) | -599.9175(3) | -599.9202(2) | **-599.9205(1)** | |
| $K^+$ (Ha) | -599.7561(2) | **-599.7679(2)** | -599.7578(2) | -599.7601(2) | -599.7589(1) | -599.7617(1) | |
| $IP_K$ (eV) | 4.278(9) | 4.000(9) | 4.286(8) | 4.28(1) | 4.388(6) | **4.321(5)** | 4.32631 |
| Fe (Ha) | -1263.6363(5) | -1263.6381(4) | -1263.6464(5) | -1263.6385(5) | -1263.6595(3) | **-1263.6613(3)** | |
| $Fe^+$ (Ha) | -1263.3570(4) | -1263.3616(4) | -1263.3620(4) | -1263.3676(4) | -1263.3746(3) | **-1263.3768(3)** | |
| $IP_{Fe}$ (eV) | 7.60(2) | 7.52(2) | 7.74(2) | 7.37(2) | **7.75(1)** | 7.74(1) | 7.84194 |
| Zn (Ha) | -1779.4101(7) | -1779.4131(6) | -1779.4195(6) | -1779.4267(5) | -1779.4304(4) | **-1779.4365(5)** | |
| $Zn^+$ (Ha) | -1779.0778(6) | -1779.0846(6) | -1779.0843(7) | -1779.0909(5) | -1779.102(1) | **-1779.1054(3)** | |
| $IP_{Zn}$ (eV) | 9.04(2) | 8.94(2) | 9.12(2) | **9.14(2)** | 8.94(3) | 9.01(2) | 9.24695 |

Table 1: Energies of third-row neutral atoms and cations. Ionization potentials are compared against experimental results from Koga et al. (1997). Total energies are in Hartree while ionization potentials are in eV. Chemical accuracy is 0.043 eV.

To investigate whether the Psiformer performance could be reproduced by simply making the FermiNet larger, we investigated both a "small" and "large" configuration for the Psiformer and FermiNet. The small FermiNet has the same layer dimensions and determinants as the one used in Pfau et al. (2020), while the large configuration has twice as many determinants and a one-electron stream twice as wide, similar to the largest networks in Spencer et al. (2020a). The Psiformer configurations have the same number of determinants and the MLP layers are the same width as the FermiNet one-electron stream, though due to the self-attention weights the small Psiformer has a number of parameters between that of the large and small FermiNet. Exact details are given in Table 7 in the appendix. On these systems, the small Psiformer ran in a similar amount of time to the large FermiNet, though for even larger systems, the difference in wall time between the small FermiNet and small Psiformer became much smaller (see Table 9 in the appendix).

The results on small molecules can be seen in Fig. 2. While increasing the size of the FermiNet increases the accuracy somewhat, the small Psiformer is more accurate than the large FermiNet, despite having fewer parameters, while the large Psiformer is the most accurate of all. This is true for all systems investigated. The improvement from the Psiformer is particularly dramatic on ozone and bicyclobutane – on ozone, the large Psiformer is within 1 kcal/mol of CCSD(T)/CBS, while even the largest FermiNet has an error more than 4 times larger than this. On all molecules, the large Psiformer captures more than 99% of the correlation energy relative to the reference energy.

## 4.2 Comparison Against FermiNet with SchNet-like Convolutions

While the performance of the Psiformer relative to the FermiNet is impressive, recent work has proposed several innovations to reach even lower energies (Gerard et al., 2022). The primary innovations of this work were the FermiNet+SchNet architecture and new hyperparameters they claim led to much faster optimization. They showed an especially large improvement on heavy atoms like potassium (K) and iron (Fe), and some improvement on larger molecules like benzene.

We attempted to reproduce the results of Gerard et al. (2022) with our own FermiNet+SchNet implementation, with somewhat surprising results in Fig. 3. First, the changes in training used for the FermiNet seem to be enough to close the gap with the published FermiNet+SchNet results on heavy atoms. For instance, ablation studies suggested that modifying the hyperparameters plus adding SchNet-like convolutions accounts for a 38.9 mHa improvement in accuracy on the potassium atom at $10^5$ iterations, but in our experiments the FermiNet with default hyperparameters is within a few mHa of the published result. On benzene, both the SchNet-like convolutions and modified hyperparameters improved the final energy by a few mHa, though still fell slightly short of the published results. Most importantly, the Psiformer is clearly either comparable to the best published FermiNet+SchNet results, as on K, or better by a wide margin, as on benzene. These results also give us confidence that the FermiNet is as strong a baseline as any other published method for further comparison.

## 4.3 Third-Row Atoms

The FermiNet, FermiNet+SchNet and Psiformer all seem to perform well on the potassium and iron atoms, but it is difficult to judge how close to the ground truth these results are. Prior work

| System | Ours | | | Best Published | |
| --- | --- | --- | --- | --- | --- |
| | FermiNet | Psiformer | | VMC | DMC |
| | | No LayerNorm | LayerNorm | | |
| Benzene | -232.2205(2) | **-232.2400(1)** | -232.2393(1) | -232.2267[a] | -232.2370(3)[b] |
| Toluene | -271.5274(2) | -271.5494(1) | **-271.5538(1)** | – | – |
| Naphthalene | -385.8147(4) | -385.8679(2) | **-385.8685(2)** | – | – |
| $CCl_4$ | -1878.684(1) | -1878.734(1) | **-1878.804(1)** | – | – |

Table 2: Energies for molecules with between 42 and 74 electrons. For benzene, the best published results using neural network Ansatzes are either from [a]Gerard et al. (2022) or [b]Ren et al. (2022).

had shown that the FermiNet can achieve energies within chemical accuracy of exact results for atoms up to argon (Spencer et al., 2020a), but exact calculations are impractical for third-row atoms. Instead, comparison to experimental results is a more practical evaluation metric. The ionization potential – the amount of energy it takes to remove one electron – is a particularly simple comparison for which good experimental data exists (Koga et al., 1997). Here we compare the FermiNet, FermiNet+SchNet and Psiformer for estimating the ionization potential of potassium, iron and zinc. To compare the relative importance of size and architecture, we also looked at both a small and large configuration of all Ansatzes, with parameters described in Table 6 the appendix.

Results are shown in Table 1. For atoms this heavy, all methods showed some amount of run-to-run variability, usually small, but on occasion as large as 10 mHa, which may explain some outlier results. On potassium, the difference between Ansatzes was within the range of run-to-run variability, but the Psiformer still did quite well, and reached the most accurate ionization potential. On heavier atoms, the difference between architectures became more pronounced, and the improvement of the Psiformer relative to other models on absolute energy was more robust. The results on ionization potentials were more mixed, and no Ansatz came within chemical accuracy of the ground truth. This shows that even the Psiformer is not yet converged to the ground truth, though it is the Ansatz closest to reaching it so far.

## 4.4 LARGER MOLECULES

Much of the promise for deep neural networks for QMC comes from the fact that they can, in theory, scale much better than other all-electron methods, though this promise has yet to be realized. While CCSD(T) scales with number of electrons as $\mathcal{O}(N^7)$, a single iteration of wavefunction optimization for a fixed neural network size scales as $\mathcal{O}(N^4)$ in theory, and in practice is closer to cubic for system sizes of several dozen electrons. Self-attention has been especially powerful when scaling to extremely large problems in machine learning – here we investigate whether the same holds true for QMC, by applying both the FermiNet and Psiformer to single molecules much larger than those which have been investigated in most prior work.

Results on systems from benzene (42 electrons) to carbon tetrachloride ($CCl_4$, 74 electrons) are given in Table 2. On these systems, CCSD(T) becomes impractical for us to run without approximations, so we only compare against other QMC results. Due to computational constraints, we were only able to run the small network configurations on these systems. Additionally, to help MCMC convergence, we updated half the electron positions at a time in each MCMC move, rather than moving all electrons simultaneously, as all-electron moves become less efficient for larger systems.

In Table 2, it can be seen that the Psiformer is not only significantly better on benzene than the FermiNet, as in Fig. 3, but it is better than the best previously published *DMC* energy, a remarkable feat. For even larger molecules, there are no results in the literature with neural network wavefunctions to compare against, so we only compare the Psiformer and FermiNet directly. The Psiformer outperforms the FermiNet by an ever larger margin on larger systems, reaching 120 mHa (75 kcal/mol) on $CCl_4$. On the three hydrocarbon systems investigated, LayerNorm has only a small impact. However for $CCl_4$ LayerNorm has a signficant impact, accounting for 70 mHa of the total 120 mHa improvement over the FermiNet. While we do not claim these are the best variational results in the literature, it is clear that on larger molecules the Psiformer is a significant improvement over the FermiNet, which is itself the most accurate Ansatz for many smaller systems.

|  | Ours | | | Ren et al. (2022) | | |
|---|---|---|---|---|---|---|
|  | FermiNet | Psiformer | | VMC | DMC | Expt. |
|  |  | No LayerNorm | LayerNorm |  |  |  |
| Equilibrium | -464.3770(5) | -464.4624(2) | **-464.4667(2)** | -464.4067(3) | –464.4640(2) |  |
| Dissociated | -464.3724(6) | **-464.4674(2)** | -464.4660(2) |  |  |  |
| $\Delta E_{\mathrm{mono}}$ | -0.0640(6) | -0.0176(2) | -0.0119(2) | -0.0219(3) | **-0.0020(5)** | 0.0038(6) |
| $\Delta E_{10\text{Å}}$ | **0.0046(8)** | -0.0050(3) | 0.0007(3) | 0.0183(5) | 0.0092(4) | 0.0038(6) |

Table 3: Energies for the benzene dimer, with center separation of 4.95 Å (equilibrium) and 10 Å (dissociated), and the estimated dissociation energy from taking the difference of the equilibrium energy from twice the mononer energy ($\Delta E_{\mathrm{mono}}$) and the dissociated energy ($\Delta E_{10\text{Å}}$). All $\Delta E_{\mathrm{mono}}$ results are based on comparing like-for-like dimer and monomer calculations. $\Delta E_{10\text{Å}}$ results from Ren et al. (2022) are estimated from figures. Experimental energies are from Grover et al. (1987).

## 4.5 THE BENZENE DIMER

Finally, we look at the benzene dimer, a challenging benchmark system for computational chemistry due to the weak van der Waals force between the two molecules, and the largest molecular system ever investigated using neural network Ansatzes (Ren et al., 2022). The dimer has several possible equilibrium configurations, many of which have nearly equal energy (Sorella et al., 2007; Azadi & Cohen, 2015), making it additionally challenging to study computationally, but here we restrict our attention to the T-shaped structure (Fig. 4) so that we can directly compare against Ren et al. (2022).

Results are given in Table 3. The results from Ren et al. (2022) are from a small FermiNet (3 layers) trained for 2 million iterations. Even our "small" 4-layer FermiNet baseline (which is larger than their FermiNet) trained for 200,000 iterations is able to reach the same accuracy as their small FermiNet trained for 800,000 iterations (Ren et al. (2022) Supplementary Figure 7a). The Psiformer reaches a significantly lower energy than the FermiNet with the same number of training iterations, again surpassing the DMC result by a few mHa.

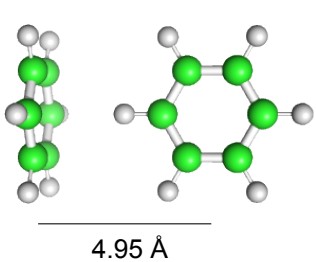

4.95 Å

Figure 4: The T-shaped benzene dimer equilibrium.

We also tried to estimate the dissociation energy by similar means as Ren et al. (2022) – comparing against the energy of the same model trained with a bond length of 10 Å, and twice the energy of the same model trained on the monomer. Ironically, our FermiNet baseline, which had the worst *absolute* energy, had the best *relative* energy between configurations. While every model underestimated the energy relative to twice the monomer, the discrepancy was lowest with DMC. It should be noted that the zero-point vibrational energy (ZPE) is not included, the dissociation energies from Ren et al. (2022) in Table 3 are estimates based on their figures, and there is disagreement over the exact experimental energy (Grover et al., 1987; Krause et al., 1991), so this comparison should be considered a rough estimate only. Our main result is that the *absolute* energy of the Psiformer is a vast improvement over the FermiNet, and we leave it to future work to properly apply the Psiformer to predicting binding energies.

## 5 DISCUSSION

We have shown that self-attention networks are capable of learning quantum mechanical properties of electrons far more effectively than comparable methods. The advantage of self-attention networks seems to become most pronounced on large systems, suggesting that these models should be the focus of future efforts to scale to even larger systems. In addition to the strong empirical results, using standard attention layers means that we can leverage existing work on improving scalability, either by architectural advances (Child et al., 2019; Wang et al., 2020; Xiong et al., 2021; Jaegle et al., 2021a;b) or software implementations optimized for specialized hardware (Dao et al., 2022), which could make it possible to scale these models even further. This presents a promising path towards studying the most challenging molecules and materials *in silico* with unprecedented accuracy.

ETHICS STATEMENT

The work presented here focuses on fundamental questions in computational chemistry, and is not yet at the stage where it is likely to be widely adopted by experimental chemists. However, in the future, this line of work could lead to computational chemistry becoming much more accurate, making it easier for chemists and material scientists to make new discoveries without requiring cumbersome trial-and-error physical experiments. This could lead to the discovery of new beneficial drugs or industrial chemical processes which are more environmentally friendly. The field of experimental chemistry already has robust ethical standards and processes for preventing harmful applications, and we are confident that more accurate computational methods will not in any way make it easier to circumvent these safeguards.

REPRODUCIBILITY STATEMENT

Details of training, network parameters, and optimization hyperparameters are given in the appendices for all experiments shown. The code is available under the Apache License 2.0 as part of the FermiNet repo at `https://github.com/deepmind/ferminet`.

ACKNOWLEDGMENTS

We would like to thank Alex G. de G. Matthews for suggesting the model name, Alex Botev for assistance with KFAC, Michael Scherbela and Leon Gerard for providing data for figures, and James Kirkpatrick for support and encouragement.

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

## A    APPENDIX

### A.1    CALCULATING ENERGIES AND GRADIENTS

It is generally more numerically stable to work directly with the log wavefunction, and the local energy can be expressed as

$$E_L(\boldsymbol{x}) = -\frac{1}{2} \sum_{i=1}^{N} \sum_{j=1}^{3} \left[ \frac{\partial^2 \log|\Psi(\boldsymbol{x})|}{\partial r_{ij}^2} + \left( \frac{\partial \log|\Psi(\boldsymbol{x})|}{\partial r_{ij}} \right)^2 \right] + V(\boldsymbol{x}) \tag{10}$$

where $V(\boldsymbol{x})$ is the potential energy (the last three terms of Eq 1).

The gradient of the energy is given by

$$\nabla \mathbb{E}_{\boldsymbol{x} \sim \Psi^2} [E_L(\boldsymbol{x})] = 2 \mathbb{E}_{\boldsymbol{x} \sim \Psi^2} \left[ (E_L(\boldsymbol{x}) - \mathbb{E}_{\boldsymbol{x}' \sim \Psi^2}[E_L(\boldsymbol{x}')]) \nabla \log|\Psi(\boldsymbol{x})| \right] \tag{11}$$

where the local energy $E_L(\boldsymbol{x}) = \Psi^{-1}(\boldsymbol{x}) \hat{H} \Psi(\boldsymbol{x})$. Note that the $E_L(\boldsymbol{x}) - \mathbb{E}_{\boldsymbol{x}' \sim \Psi^2}[E_L(\boldsymbol{x}')]$ term is the difference between the local energy at $\boldsymbol{x}$ and the average energy over all $\boldsymbol{x}'$.

We make a small but critical change to how the gradients are computed relative to Pfau et al. (2020) which stabilizes training for *all* models considered here. The local energy often has very large tails due to numerical issues, especially near cusps (where two particles overlap) and nodes (where the wavefunction goes to zero) (Pathak & Wagner, 2020). To mitigate this, the local energy is often truncated in practice. Let $\langle E_L \rangle_{\mathrm{mean}}$ denote the mean local energy for one minibatch of walkers and $\langle E_L \rangle_{\mathrm{median}}$ denote the median. Then in Pfau et al. (2020), the local energies were clipped to be within a constant multiple $\rho$ of the mean absolute deviation around the mean: $\mathrm{MAD}_{\mathrm{mean}}(E_L) = \frac{1}{N} \sum_i |E_L(\boldsymbol{x}_i) - \langle E_L \rangle_{\mathrm{mean}}|$. While this is more robust than the standard deviation, it is still susceptible to large outliers because it is centered at the mean. Instead, we use the mean absolute deviation around the *median* to determine the window for clipping: $\mathrm{MAD}_{\mathrm{median}}(E_L) = \frac{1}{N} \sum_i |E_L(\boldsymbol{x}_i) - \langle E_L \rangle_{\mathrm{median}}|$.

Secondly, in Pfau et al. (2020), the average energy term $\mathbb{E}_{\boldsymbol{x}' \sim \Psi^2}[E_L(\boldsymbol{x}')]$ in the gradient was approximated by the mean local energy $\langle E_L \rangle_{\mathrm{mean}}$ of a minibatch. That meant that outliers were still included in this term. We instead use the mean of the *clipped* local energies. This guarantees that the mean over a minibatch of the energy difference term is always zero, improving stability during optimization. If we let $\mathrm{clip}_{\mathrm{mean/median}}(x; \sigma)$ denote the functions that clip $x$ to be in the range $[\langle x \rangle_{\mathrm{mean/median}} - \sigma, \langle x \rangle_{\mathrm{mean/median}} + \sigma]$, then the gradient for one batch in Pfau et al. (2020) was:

$$\hat{E}_L(\boldsymbol{x}) = \mathrm{clip}_{\mathrm{mean}}(E_L(\boldsymbol{x}); \rho \mathrm{MAD}_{\mathrm{mean}}(E_L)) \tag{12}$$

$$\left\langle \left( \hat{E}_L(\boldsymbol{x}) - \langle E_L \rangle_{\mathrm{mean}} \right) \nabla \log|\Psi(\boldsymbol{x})| \right\rangle_{\mathrm{mean}} \tag{13}$$

while here we use:

$$\hat{E}_L(\boldsymbol{x}) = \mathrm{clip}_{\mathrm{median}}(E_L(\boldsymbol{x}); \rho \mathrm{MAD}_{\mathrm{median}}(E_L)) \tag{14}$$

$$\left\langle \left( \hat{E}_L(\boldsymbol{x}) - \left\langle \hat{E}_L \right\rangle_{\mathrm{mean}} \right) \nabla \log|\Psi(\boldsymbol{x})| \right\rangle_{\mathrm{mean}} \tag{15}$$

While this may seem like a subtle difference, it has a great effect on larger systems, especially heavier atoms. A similar technique was used to stabilize training the FermiNet with pseudopotentials (Li et al., 2022), but instead of clipping outlier local energies, the outlier walkers were removed entirely for that minibatch. We found that clipping with proper centering was more effective than removing the walkers entirely.

### A.2    TRAINING

In this section we give details on training and further differences from the original FermiNet (Pfau et al., 2020).

#### A.2.1    DENSE DETERMINANTS

The original FermiNet and PauliNet Ansatzes used block-diagonal determinants for spin-up and spin-down electrons (Pfau et al., 2020; Hermann et al., 2020), such that each determinant could be

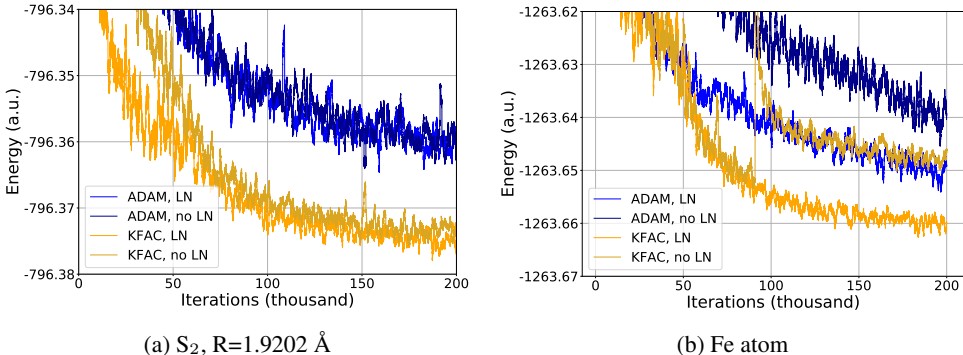

(a) $S_2$, R=1.9202 Å

(b) Fe atom

Figure 5: Comparison of different optimization algorithms (KFAC and ADAM) for the Psiformer on (a) the sulphur dimer and (b) the iron atom, with and without LayerNorm. While there is a clear advantage to using KFAC on both systems, LayerNorm has a marginal impact on the sulphur dimer. On the iron atom, however, LayerNorm improves the accuracy of ADAM and the stability of KFAC. A learning rate of 3e-4 was used for ADAM. A rolling mean of the last 1000 iterations is used to smooth the energy.

factorized into the product of a determinant of spin-up electrons of size $N_\uparrow \times N_\uparrow$ and a determinant of spin-down electrons $N_\downarrow \times N_\downarrow$, as is the case with conventional VMC ansatzes (Foulkes et al., 2001):

$$\det\left[\mathbf{\Phi}(\boldsymbol{x})\right] = \begin{vmatrix} \mathbf{\Phi}_\uparrow(\boldsymbol{x}_\uparrow) & \mathbf{0} \\ \mathbf{0} & \mathbf{\Phi}_\downarrow(\boldsymbol{x}_\downarrow) \end{vmatrix} = \det\left[\mathbf{\Phi}_\uparrow(\boldsymbol{x}_\uparrow)\right]\det\left[\mathbf{\Phi}_\downarrow(\boldsymbol{x}_\downarrow)\right]. \tag{16}$$

where $\boldsymbol{x}_\sigma$ denotes the set of electron states with the specified spin and different functions are used for electrons of different spins. The FermiNet authors subsequently proposed simply using dense determinants of size $N \times N$, $N = N_\uparrow + N_\downarrow$ (Spencer et al., 2020b):

$$\det\left[\mathbf{\Phi}(\boldsymbol{x})\right] = \left|\mathbf{\Phi}_\uparrow(\boldsymbol{x}_\uparrow) \quad \mathbf{\Phi}_\downarrow(\boldsymbol{x}_\downarrow)\right| \tag{17}$$

where now each block is of dimension $N \times N_\sigma$. This has largely become the default choice for FermiNet architectures and has shown to provide improved accuracy at small additional cost for a myriad of systems (Lin et al., 2021; Cassella et al., 2022; Ren et al., 2022; Gerard et al., 2022; Gao & Günnemann, 2022).

### A.2.2 IMPLEMENTATION OF FERMINET+SCHNET

To best reproduce the results from Gerard et al. (2022), we implemented the changes to the FermiNet which they claimed made the largest difference. All experiments in this paper, including the FermiNet, used dense determinants. We added the SchNet-like continuous filter convolutions, as well as the nuclear embedding stream and separate streams for same-spin and different-spin electrons in the two-electron stream. We also compared their training hyperparameters against ours, except for batch size, which was kept at 4096 for all experiments. We did not implement their local input features or envelope initialization, as their results suggested they did not make as significant a difference, and our longer pretraining likely had the same effect as changing the envelope initialization.

### A.2.3 CHOICE OF OPTIMIZER AND LAYERNORM

The original FermiNet was only able to reach high accuracy when trained with Kronecker-factored approximate curvature (KFAC) (Martens & Grosse, 2015). To see if the same holds true for the Psiformer, here we compare training with KFAC against ADAM on several systems. Additionally, when trained with ADAM, self-attention layers usually require LayerNorm (Ba et al., 2016). However, other work has suggested that in some contexts normalization may not be necessary when using KFAC (Martens et al., 2021), so we also compare the Psiformer with and without LayerNorm. Figure 5 shows a comparison for the Psiformer on the sulphur dimer and the iron atom. Consistent with the FermiNet, the Psiformer consistently converges faster and to lower energies with KFAC. With LayerNorm, the situation is more ambiguous. On the sulphur dimer, it seems to have marginal

|  | Parameter | Value |
|---|---|---|
| Training | Training iterations | 2e5 |
|  | Learning rate at time $t$ | $lr_0(1 + \frac{t}{t_0})^{-1}$ |
|  | Initial learning rate $lr_0$ | 0.05 |
|  | Learning rate decay $t_0$ | 1e5 |
|  | Local energy clipping $\rho$ | 5.0 |
| Pretraining | Pretraining optimizer | LAMB |
|  | Pretraining iterations | 2e4 or 1e5 |
|  | Pretraining basis set | STO-6G |
| Markov Chain Monte Carlo | Batch size | 4096 |
|  | Decorrelation steps | 30 |
| KFAC | Norm constraint | 1e-3 |
|  | Damping | 1e-3 |

Table 4: Table of default hyperparameters used.

| Systems | Pretraining Steps | MCMC Blocks | LayerNorm |
|---|---|---|---|
| Small molecules (Fig. 2) | 20,000 | 1 | No |
| Third-row atoms (Table 1) | 100,000 | 2 | Yes |
| Large molecules (Table 2) | 100,000 | 2 | Both |
| Benzene dimer | 100,000 | 4 | Both |

Table 5: Variations in hyperparameters between experiments.

impact, while on the iron atom, it definitely improves training with ADAM, and seems to improve stability when using KFAC.

### A.2.4 HYPERPARAMETERS

Table 4 shows the default hyperparameters used for training all models implemented in this work. Note that Pfau et al. (2020) took the sum over gradients across the batch on each device and averaged over devices, whereas here the gradients are averaged over the entire batch. This amounts to a scaling of the learning rate, meaning that the same learning rate as in Pfau et al. (2020) cannot be used.

As in previous FermiNet implementations, we pretrain all networks to match Hartree-Fock (HF) orbitals computed using PySCF. Here, we use the LAMB optimizer (You et al., 2020) as the pretraining optimizer. We find that longer pretraining stabilises training for all models considered. In addition, during pretraining, we draw samples from the HF orbitals only, instead of the neural network wavefunction. A smaller number of pretraining iterations (20,000) was used for small molecules in Section 4.1, while 100,000 iterations were used for third-row atoms and larger molecules.

To generate samples from $\Psi^2$, we use the Metropolis-Hastings algorithm with symmetric Gaussian proposals, as in Pfau et al. (2020). We increase the number of decorrelation MCMC steps between optimization iterations from 10 in Pfau et al. (2020) to 30. Additionally, for larger systems, we do not update all electron positions simultaneously in one Metropolis step. Instead, we split the electrons into multiple blocks, and iteratively update each block once per step. This is an intermediate between the all-electron and one-electron moves commonly used in VMC.

Note that while all models were trained for 200,000 optimization iterations, this does not mean that all systems required that many iterations to converge. Many smaller systems converged in far fewer iterations than this, but the same number was used for all systems to minimize confusion.

Not all systems used identical parameters - for larger systems we increased the number of pretraining steps and blocks per MCMC update, and LayerNorm was not used for small molecules in Table 2. In Table 5 we specify which systems were trained with which settings.

In Section 4.1, the performance of "small" and "large" FermiNet and Psiformer models was investigated on a set of small and medium molecules. Table 6 gives the network parameters used for these

| Parameter | FermiNet | | Psiformer | |
|---|---|---|---|---|
| | Small | Large | Small | Large |
| Determinants | 16 | 32 | 16 | 32 |
| Network layers | 4 | 4 | 4 | 4 |
| Attention heads | N/A | N/A | 4 | 8 |
| Attention dimension | N/A | N/A | 64 | 64 |
| MLP hidden dims: one-electron | 256 | 512 | 256 | 512 |
| MLP hidden dims: two-electron | 32 | 32 | N/A | N/A |

Table 6: Table of network parameters used for the Psiformer and FermiNet.

| System | | FermiNet | | | Psiformer | |
|---|---|---|---|---|---|---|
| | | Pfau et al. (2020) | Small | Large | Small | Large |
| LiH | Energy | -8.0705(1) | -8.070050(1) | -8.070515(8) | -8.070528(5) | -8.070536(4) |
| | # parameters | 668128 | 683744 | 2610400 | 1593858 | 6366210 |
| Li2 | Energy | -14.99475(1) | -14.99480(2) | -14.99484(2) | -14.99486(1) | -14.99485(2) |
| | # parameters | 676960 | 700384 | 2676448 | 1602178 | 6399234 |
| NH3 | Energy | -56.56295(8) | -56.56347(4) | -56.56387(3) | -56.56367(2) | -56.56381(2) |
| | # parameters | 703968 | 741088 | 2823392 | 1621506 | 6470658 |
| CH4 | Energy | -40.51400(7) | -40.51430(3) | -40.51450(3) | -40.51454(2) | -40.51461(1) |
| | # parameters | 708640 | 744800 | 2830816 | 1622850 | 6473346 |
| CO | Energy | -113.3218(1) | -113.32354(7) | -113.32444(5) | -113.32416(4) | -113.32466(3) |
| | # parameters | 712288 | 766944 | 2940640 | 1635458 | 6531330 |
| N2 | Energy | -109.5388(1) | -109.54046(6) | -109.54148(6) | -109.54137(4) | -109.54179(4) |
| | # parameters | 712288 | 766944 | 2940640 | 1635458 | 6531330 |
| C2H4 | Energy | -78.5844(1) | -78.58604(5) | -78.58701(6) | -78.58762(3) | -78.58794(3) |
| | # parameters | 743648 | 799968 | 3039456 | 1649922 | 6576642 |
| methylamine | Energy | -95.8554(2) | -95.85917(6) | -95.86010(5) | -95.86050(4) | -95.86096(3) |
| | # parameters | 759712 | 821344 | 3114976 | 1660098 | 6613378 |
| O3 | Energy | -225.4145(3) | -225.4226(2) | -225.4268(1) | -225.43061(9) | -225.43231(8) |
| | # parameters | 763360 | 854752 | 3280096 | 1678850 | 6700034 |
| ethanol | Energy | -155.0308(3) | -155.0419(1) | -155.0455(1) | -155.04656(7) | -155.04759(6) |
| | # parameters | 815904 | 899936 | 3403232 | 1698370 | 6755458 |
| bicyclobutane | Energy | -155.9263(6) | -155.9388(1) | -155.9432(1) | -155.94619(8) | -155.94836(7) |
| | # parameters | 845920 | 940000 | 3548896 | 1717890 | 6827266 |

Table 7: Energies and number of parameters in network for set of small molecules studied in Pfau et al. (2020). Note Pfau et al. (2020) used block diagonal determinants and did not report precise parameter counts; parameters counts for their results are estimated using their published settings and the FermiNet JAX implementation (Spencer et al., 2020b). Our FermiNet experiments used the envelope function proposed in Spencer et al. (2020a) and dense determinants, resulting in slightly different numbers of parameters between the networks of Pfau et al. (2020) and our FermiNet (Small) networks.

models. In all other experiments, unless otherwise specified, the 'small' network configurations were used. Table 7 contains the number of parameters used by each model for this set of molecules.

## A.3 COMPUTATIONAL DETAILS

All models were implemented in JAX (Bradbury et al., 2018) based upon the public FermiNet (Spencer et al., 2020b) and KFAC implementations (Botev & Martens, 2022), and trained in parallel using between 16 and 64 A100 GPUs, depending on the system size. All calculations were done at standard single precision, as we found that TensorFloat-32 calculations on A100 had numerical accuracy issues. A table of total training time – including pretraining – for several models is given in Table 9. Empirically, the time per iteration scaled roughly cubically, i.e. the benzene dimer required ~8 times the computational resources to run as the benzene molecule, though the number of atoms also played a significant role in timing. For systems with very large numbers of atoms, like the benzene dimer, the FermiNet and Psiformer ran at nearly the same speed. This is likely because the determinants and envelope, which are similar between the FermiNet and Psiformer, make up a larger share of the total run time as the systems become larger. Geometries are taken from Curtiss et al. (2000) and given in Table 8 or in Pfau et al. (2020). The benzene dimer geometry is obtained via a rigid translation and rotation of two monomers.

| System | atom | position ($a_0$) | | |
|---|---|---|---|---|
| benzene | C | 0.00000 | 2.63664 | 0.00000 |
| | C | 2.28339 | 1.31832 | 0.00000 |
| | C | 2.28339 | -1.31832 | 0.00000 |
| | C | 0.00000 | -2.63664 | 0.00000 |
| | C | -2.28339 | -1.31832 | 0.00000 |
| | C | -2.28339 | 1.31832 | 0.00000 |
| | H | 0.00000 | 4.69096 | 0.00000 |
| | H | 4.06250 | 2.34549 | 0.00000 |
| | H | 4.06250 | -2.34549 | 0.00000 |
| | H | 0.00000 | -4.69096 | 0.00000 |
| | H | -4.06250 | -2.34549 | 0.00000 |
| | H | -4.06250 | 2.34549 | 0.00000 |
| toluene | C | -0.01831 | 1.72486 | 0.00000 |
| | C | -0.01297 | 0.37035 | 2.27060 |
| | C | -0.01297 | 0.37035 | -2.27060 |
| | C | -0.01297 | -2.26452 | 2.27703 |
| | C | -0.01297 | -2.26452 | -2.27703 |
| | C | -0.00900 | -3.59169 | 0.00000 |
| | C | 0.05583 | 4.56877 | 0.00000 |
| | H | 2.00464 | 5.26559 | 0.00000 |
| | H | -0.88281 | 5.33834 | -1.67217 |
| | H | -0.88281 | 5.33834 | 1.67217 |
| | H | -0.01841 | -3.28187 | 4.06225 |
| | H | -0.01841 | -3.28187 | -4.06225 |
| | H | -0.01415 | -5.64576 | 0.00000 |
| | H | -0.02402 | 1.39270 | 4.05592 |
| | H | -0.02402 | 1.39270 | -4.05592 |
| naphthalene | C | 0.00000 | 0.00000 | 1.35203 |
| | C | 0.00000 | 0.00000 | -1.35203 |
| | C | 0.00000 | 2.34349 | 2.64930 |
| | C | 0.00000 | -2.34349 | 2.64930 |
| | C | 0.00000 | 2.34349 | -2.64930 |
| | C | 0.00000 | -2.34349 | -2.64930 |
| | C | 0.00000 | -4.59147 | 1.33509 |
| | C | 0.00000 | 4.59147 | 1.33509 |
| | H | 0.00000 | 2.34107 | 4.70689 |
| | H | 0.00000 | -2.34107 | 4.70689 |
| | C | 0.00000 | -4.59147 | -1.33509 |
| | C | 0.00000 | 4.59147 | -1.33509 |
| | H | 0.00000 | 2.34107 | -4.70689 |
| | H | 0.00000 | -2.34107 | -4.70689 |
| | H | 0.00000 | -6.37654 | 2.35261 |
| | H | 0.00000 | 6.37654 | 2.35261 |
| | H | 0.00000 | -6.37654 | -2.35261 |
| | H | 0.00000 | 6.37654 | -2.35261 |
| $CCl_4$ | C | 0.00000 | 0.00000 | 0.00000 |
| | Cl | 1.93005 | 1.93005 | 1.93005 |
| | Cl | -1.93005 | -1.93005 | 1.93005 |
| | Cl | -1.93005 | 1.93005 | -1.93005 |
| | Cl | 1.93005 | -1.93005 | -1.93005 |

Table 8: Molecular geometries in Bohr.

| System | FermiNet | | FermiNet+SchNet | | Psiformer | |
|---|---|---|---|---|---|---|
| | Small | Large | Small | Large | Small | Large |
| Bicyclobutane | 484 | 789 | – | – | 712 | 1412 |
| Zn | 592 | 1200 | 808 | 1365 | 1317 | 1600 |
| Benzene | 1224 | – | 1608 | – | 1768 | – |
| Benzene Dimer | 10576 | – | – | – | 10695 | – |

Table 9: Number of A100 GPU hours required to train several systems.

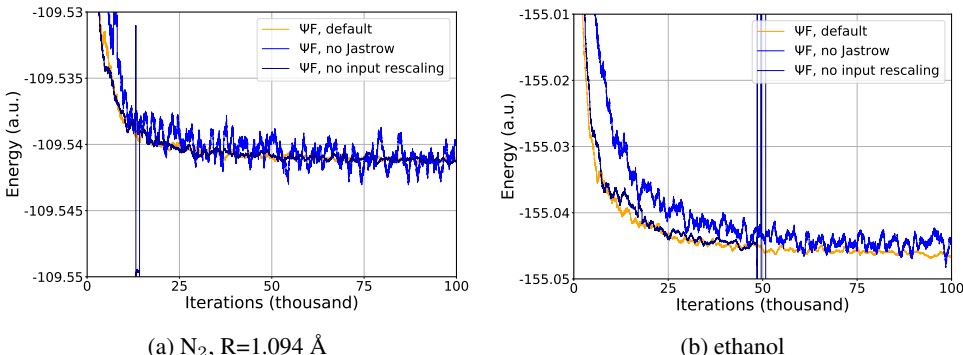

(a) $N_2$, R=1.094 Å

(b) ethanol

Figure 6: Ablation studies for the Psiformer without a Jastrow factor or input rescaling on (a) the nitrogen dimer and (b) ethanol. A rolling mean of the last 1000 iterations is used to smooth the energy. The divergence of the energy without using input rescaling, as shown here for ethanol, is typical for medium or large systems.

## A.4 ABLATION STUDIES

Two modifications to the self-attention network proved critical to stability for the Psiformer - the Jastrow factor ensuring correct behaviour at electronic cusps, and rescaling the input distances. Here we show ablation studies without these modifications.

Figure 6 shows the Psiformer on (a) the nitrogen dimer and (b) ethanol, without a Jastrow factor and without input feature rescaling. In the absence of an electron-electron Jastrow factor to enforce cusp conditions, the energy is very noisy. If the input features are not rescaled, the energy is often unstable. For smaller systems, as in the nitrogen dimer shown, training may recover from the instability, but especially for medium or larger systems, the energy often diverges.

## A.5 ATTENTION MAPS

Figures 7 and 8 show attention maps of the Psiformer for a sample electron configuration, for the benzene dimer at equilibrium and dissociated geometry respectively. Each attention map shows the output of the dot product between key and query vectors before the softmax operation. Four attention maps are shown for each network layer, one per attention head. The ordering of the electrons is chosen to highlight the structure of the attention maps in relation to electron distances: the electrons are grouped according to their closest atom, where the atoms of each benzene molecule are ordered around the ring. The block structure of the attention weights is clearly visible, where electrons attend more to other electrons around the same molecule.

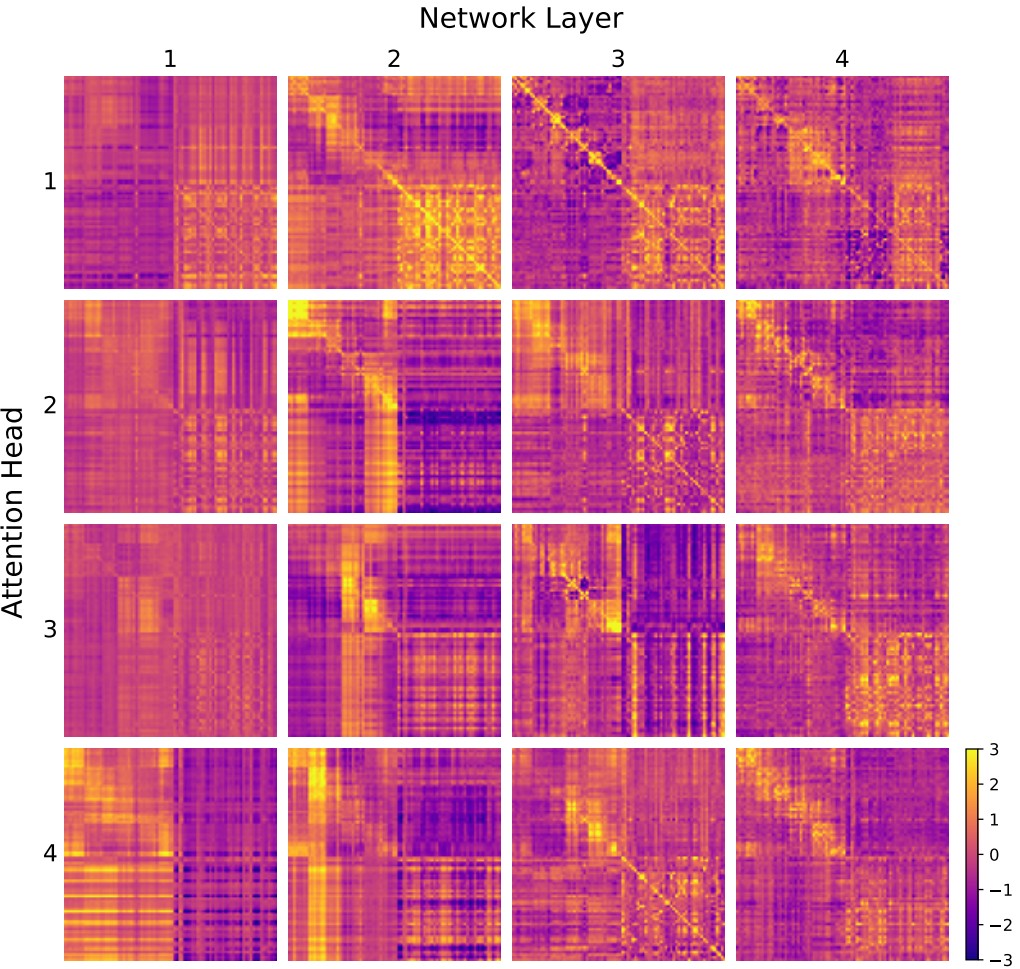

Figure 7: Attention maps for the Psiformer for a sample electron configuration from the benzene dimer at equilibrium geometry. For each network layer, the attention maps for 4 attention heads are shown. The electrons are grouped by nearest atom.

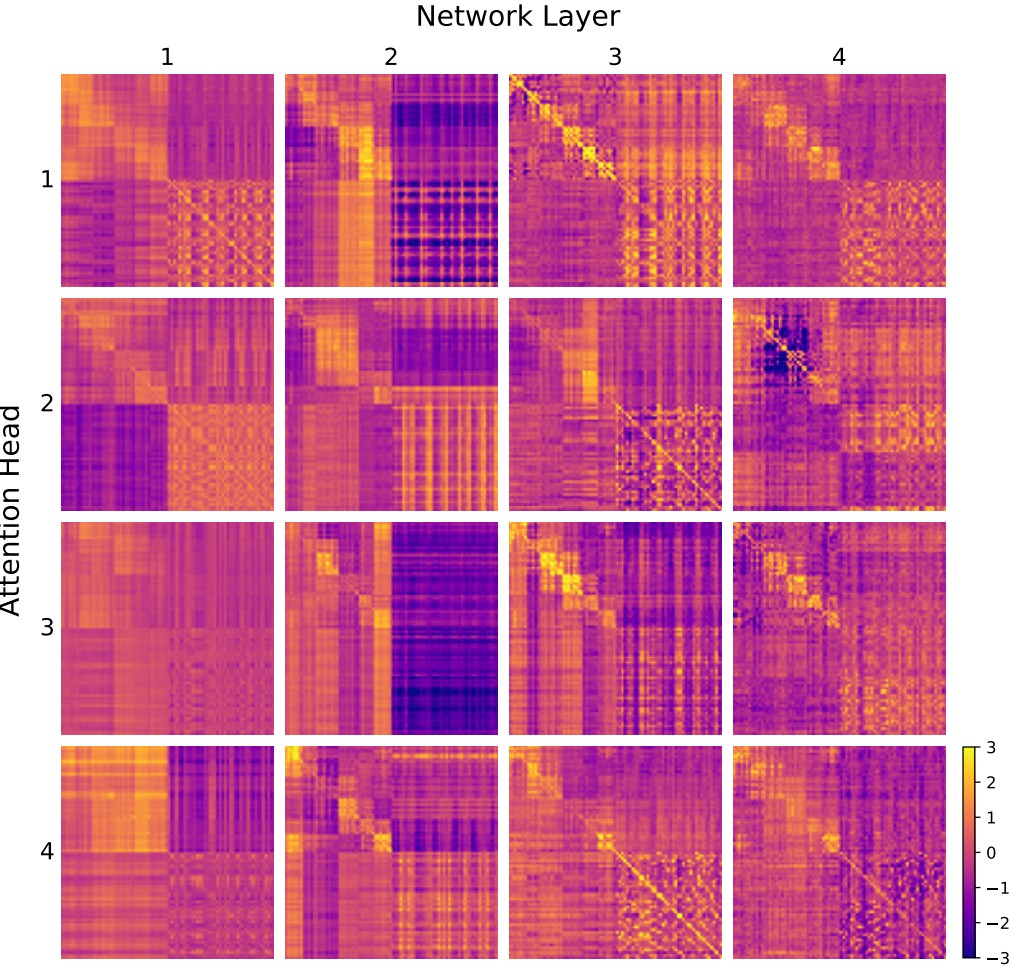

Figure 8: Attention maps for the Psiformer for a sample electron configuration from the benzene dimer at dissociated geometry. For each network layer, the attention maps for 4 attention heads are shown. The electrons are grouped by nearest atom.

