# OpenReview forum: "A Self-Attention Ansatz for Ab-initio Quantum Chemistry"
_ICLR.cc/2023/Conference — ICLR 2023 poster_

### Official Review · Reviewer_GTMe · 2022-10-20

**Confidence:** 4
**Clarity, Quality, Novelty And Reproducibility:** The paper has good clarity, quality, …
**Correctness:** 4
**Technical Novelty And Significance:** 3
**Empirical Novelty And Significance:** 3
**Recommendation:** 8

**Strength And Weaknesses:**

Strength: the main strength of this paper is that --- it really works. I believe there are many researchers actively working on incorporating complicated model structure in FermiNet, and many of them may have the idea to use self-attention. This paper is the first one to make this work, which is not a naive accomplishment. The runtime scaled cubically, which means for large systems, FermiNet and PsiFormer ran at nearly the same speed. Also the experiments are very detailed.

Weaknesses: I don't see any major weaknesses, but I have several questions or comments below.

1. Why do you only consider features of electrons? Can we include the features of atoms in the self-attention mechanism? (DeepWF https://arxiv.org/pdf/1807.07014.pdf did something like this.)
2. More ablation study would be helpful for understanding how the number of determinants, number of attention heads, attention dimensions and etc (basically the hyperparameters listed in table 6) make contributions to the accuracy of calculation.
3. The divergence of training energy as in Figure 6 without input rescaling seems quite outrageous. Does the authors see similar issue in FermiNet and FermiNet SchNet?
4. The authors claim that this work is the first time that KFAC is applied to self-attention layer (in Section 4). However, https://arxiv.org/abs/1907.04164 (from KFAC authors) already claimed that this is the first work where KFAC is applied to transformer model which include self-attention.

**Summary Of The Paper:**

This paper proposed to use self-attention as ansatz in FermiNet. There are some minor changes, including using log distance and spin in the input feature. The proposed model is proved better than FermiNet in several experiments including third-row atoms, large molecules and benzene dimer.

**Summary Of The Review:**

This paper introduce self-attention in the ansatz in FermiNet. I guess the idea may have been studied by many researchers, and this paper is the first one to make it work, which is a good accomplishment.

---

> ### Author Response · Authors · 2022-11-08
> **Response to Reviewer GTMe**
>
> We thank the reviewer for their kind words, and for recognizing the significance of our contribution. In response to the reviewers questions:
> 1. Given the atomic positions are fixed during training, we are unclear on what including them as inputs to the network would accomplish. We believe this makes sense for models like PESNet (https://openreview.net/forum?id=apv504XsysP) and PlaNet (https://arxiv.org/abs/2205.14962), where the network learns an ansatz for multiple systems, but for learning the ground state of a single system, atomic positions are constant, and including constant inputs to a neural network should not meaningfully change what they are capable of learning. Although some methods like DeepWF and FermiNet+SchNet did use atomic positions as input, we are skeptical that this had any meaningful impact on their performance.
> 2. We are happy to include further ablation studies in the appendix. Some of the systems studied in the paper require very large computational resources to train - is there a particular system which you think would be most informative to study?
> 3. We agree that the divergence in training the PsiFormer without the log-rescaled inputs is quite extreme. This was unique to the PsiFormer and we did not see it with FermiNet or FermiNet+SchNet.
> 4. Thank you for the pointer. We have included a citation in the paper, and rephrased the discussion at the start of Sec. 4 to acknowledge their work.

---

### Official Review · Reviewer_RSBm · 2022-10-21

**Confidence:** 2
**Correctness:** 3
**Technical Novelty And Significance:** 3
**Empirical Novelty And Significance:** 3
**Recommendation:** 6

**Clarity, Quality, Novelty And Reproducibility:**

Clarity: This paper provides a nice overview of the problem being solved. There are some details missing (like the sampling algorithm for VMC), but I could understand the paper even though I come from a machine learning background.

Novelty: The novelty is not very high since self-attention is not a very new concept. However, I think this is fine given importance of the problem.

Reproducibility: Without any repository, I do not think it is possible to reproduce the paper.

**Strength And Weaknesses:**

Strength 1: This paper studies a very important problem in computational chemistry: solving the many-electron Schrodinger equation. Even a small progress in this area is likely to be impactful for future research.

Strength 2: The proposed self-attention mechanism is reasonable since the self-attention mechanism has shown success for many machine learning applications.

Strength 3: The experiments are thorough and it looks like the authors did a good job at solving a complex problem that requires a lot of engineering.

Weakness 1: The proposed idea itself is not very novel. To my knowledge, the novelty of this paper is limited to the self-attention mechanism and the Jastrow factor-based parameterization of electron-electron features. Both of them has already been considered in prior works, like the FermiNet.

Weakness 2: I hope the authors could further highlight the running time of their algorithm in the main text. I think this is importance since (1) running time is important for this type of problem and (2) the proposed self-attention mechanism is known to be quite slow. Currently, I find that the authors report their training time in Table 9 of the appendix. I guess this is the time required to solve the Schrodinger equation for a molecule. How does this time compare to the existing DNN-based or non-DNN-based solvers?

Weakness 3: I would like to have seen some justification behind using the self-attention mechanism. Besides the additional expressive power, is there any reason that self-attention is more useful for parameterization of the energy function? Is there any gating mechanism in “ground truth” for solving the Schrodinger equation?

Weakness 4: There is no explanation on which Markov chain Monte Carlo algorithm is used for the variational quantum Monte Carlo.

Weakness 5: Since the authors do not provide a code repository for their paper, given complexity of the problem being solved, the possibility of reproducing this paper’s experiment is very low.

Question: There are some works on graph-based Transformer to incorporate pairwise features to self-attention like Graphormer. I wonder if the authors have already considered this type of incorporation for their work, but discarded due to poor performance.

**Summary Of The Paper:**

This paper proposes a new attention-based neural architecture for solving many-electron Schrodinger equation by optimizing the variational quantum Monte Carlo objective. While previous works proposed architectures like FermiNet, the proposed PsiFormer differs by (1) using self-attention between $O(N)$ electron-nuclear features and (2) incorporating $O(N^{2})$ electron-electron features through (parameterized) Jastrow factor. The proposed algorithm empirically outperforms the prior works in various experiments.

**Summary Of The Review:**

This paper proposes a simple idea for solving an important problem. I believe significance and complexity of the problem being solved is enough to compensate for the (arguable) lack of novelty. I am concerned about reproducibility of this paper. I do not think the follow-up works will be able to reproduce this paper without allocating a significant amount of time.

---

> ### Author Response · Authors · 2022-11-08
> **Response to Reviewer RSBm**
>
> We thank the reviewer for their comments and kind words in regards to the strengths of the paper, and respond to their concerns below:
>
> 1. The reviewer states that both self-attention and Jastrow factors were considered in the original FermiNet. We do not believe this is accurate. While the PauliNet (https://www.nature.com/articles/s41557-020-0544-y) incorporated both electron-electron and electron-nuclear Jastrow factors, the FermiNet did not. Neither the FermiNet nor PauliNet used self-attention. The PauliNet used continuous-filter convolutions similar to the SchNet, and later work integrated this into a FermiNet-like architecture. We investigated this FermiNet+SchNet as a baseline and found it was not as accurate as the PsiFormer. Even though self-attention and Jastrow factors are not original on their own, we believe that showing that their combination achieves state-of-the-art performance on challenging quantum chemical benchmarks is an important achievement.
>
> 2. While the main text of the paper is already quite tight, we have added a short discussion of wall time in Section 4.1, with pointers to Table 9 in the appendix which already included a longer discussion of this topic. For a comparison against other DNN methods, many papers in this field include their own discussion of runtime (for instance, Appendix C of Gerard et al. 2022, or Table 2 of Spencer, Pfau, Botev and Foulkes 2020), but most models based on the FermiNet run in roughly equivalent amounts of time, while the PauliNet is faster, as it has fewer parameters. Most non-DNN VMC code (e.g. CASINO, QMCPACK, etc) is not optimized for GPU, and so direct comparison is difficult, but they generally require fewer flops than the methods presented here, as the ansatzes are even simpler. The optimization methods used for non-DNN ansatzes are also quite different - typically there is a larger number of MCMC steps between parameter updates, and only a small number of parameter updates, (perhaps a few hundred). The main advantage of the method presented here is accuracy, which no non-DNN VMC ansatz can match.
>
> 3. While it is difficult to know exactly why the self-attention layers are so effective, some reasonable physical intuition can help motivate it. Generally, the effects of electron correlation become weaker as electrons move further apart, and so it is reasonable to expect that a mechanism for gating the strength of interaction between different electrons would help improve the accuracy of any ansatz. Beyond that, it is difficult to say exactly why self-attention is so effective.
>
> 4. We mention in section A.2.4 that Metropolis (or symmetric Metropolis-Hastings) steps are used, but we have rewritten this section to make it more explicit, as well as mentioning it at the beginning of Sec. 4 in the main text of the paper.
>
> 5. As we state in the reproducibility statement, we fully intend to release our code open-source before the conference, and believe that doing so is consistent with ML conference community standards, where the code is not always ready at the time of paper submission, but instead released several months later. Unfortunately, we are not able to share the code at this time, but we have made significant progress in cleaning up the code for public release, and believe it will be ready in a few months. While we cannot prove this without de-anonymizing ourselves, we hope you will take us at our word when we say we have a perfect track record of releasing all code from our contributions to machine learning venues.
>
> We have not looked at using the Graphformer to incorporate pairwise information into the PsiFormer, and we thank the reviewer for the suggestion. The PsiFormer was the first self-attention based ansatz we found which worked, but we are open to suggestions for future directions which may improve our results even further.

---

> > ### Comment · Reviewer_RSBm · 2022-11-09
> > **Thanks for the detailed response.**
> >
> > Thank you for the clarification.
> >
> > 1. Thank you for the correction (regarding Jacobi factor-based parameterization), and sorry for the confusion (regarding self-attention). To clarify one of my points, I meant that many works have used self-attention to model pair-wise relation of data, and this paper follows that line of research. Nevertheless, I agree that achieving state-of-the-art performance on a benchmark like this is a significant result on its own.
> >
> > 2. I do not think your "General Response " accurately summarizes the overall reviewing process so far. Please note that I was not asking for details that already exist in the appendix. I read the appendix before writing my reviews and I knew that the running time and the detail of using the metropolis-hastings algorithm were included in the appendix. I was (1) asking for "giving more highlights" to the running time since it is practically important and (2) curious about which proposal distribution you used for the metropolis hastings algorithm.

---

> > > ### Author Response · Authors · 2022-11-09
> > > **Continuing the discussion**
> > >
> > > Apologies if we mischaracterized your comments in the General Response. We were trying to summarize the comments from many reviewers and may have painted things with too broad a brush. We appreciate the clarification.
> > >
> > > Thank you for reading the paper in detail, including the appendix, and apologies again if we mischaracterized your request. In our updated paper, we have extended section A.2.4 to explicitly state that a Gaussian proposal distribution is used, and that the distribution is symmetric (p(x|x') = p(x'|x)). We also added a sentence to the second paragraph of section 4.1 where we highlight the timing results in the appendix, whereas previously we did not discuss them in the main text at all. We hope that this additional detail helps clear up any ambiguity, but please let us know if there is anything else we could do to improve the paper or convince you to raise your score.

---

> > > > ### Comment · Reviewer_RSBm · 2022-11-17
> > > > **Thank you for the clarification**
> > > >
> > > > Overall I am satisfied with the response. I maintain my score as "weak accept" since I do think this paper solves a significant problem, but the novelty is moderate when compared to the vast literature on 3D graph representation learning.

---

### Official Review · Reviewer_58NZ · 2022-10-24

**Confidence:** 4
**Correctness:** 2
**Technical Novelty And Significance:** 2
**Empirical Novelty And Significance:** 2
**Recommendation:** 5

**Clarity, Quality, Novelty And Reproducibility:**

The language used in this paper is very professional, but the explanation of terms is too brief, which is not very friendly to non-major readers. For the experimental part, there are too few baselines selected, which makes the result not strong enough. It is recommended to introduce more baselines or consider adding different kinds of attention mechanisms in these frameworks to compare with PsiFormer. I believe that the results will be more convincing.

**Strength And Weaknesses:**

The idea is novel for AI+Science. However, there are some incomplete or vague expressions in this paper. Please improve the language presentation carefully. The motivation for joining self-attention is not clear enough. How does the attention mechanism work on gating interactions between electrons and make the framework better?

**Summary Of The Paper:**

This is an interdisciplinary research paper on AI for science, it proposes a neural network framework based on self-attention mechanism, which can get an Anstazes solving the many-electron Schrodinger equation. The authors scaled the input features and concatenate the numerical spin into the input feature vector. Then they project input features to the same dimension of attention inputs by linear mapping to add self-attention mechanism into neural network framework.

**Summary Of The Review:**

AI for science is novel for our community. However, the contribution of this work is to introduce self-attention. It is an incremental contribution.

---

> ### Author Response · Authors · 2022-11-08
> **Response to Reviewer 58NZ**
>
> We thank the reviewer for their comments, and hope that we can address their comments to demonstrate the novelty and significance of our work. We apologize if some of the wording was vague or unclear. However, the reviewer’s comments are very brief and lacking in detail. Could you provide us some examples of parts which you found unclear or in need of improvement? Without more detailed feedback, we are not able to make the paper better. We also apologize if the description of some terms is very brief, but ICLR does have a strict policy on page limits, and there is a large volume of material to fit into the paper to make it accessible to both machine learning and quantum chemistry communities. We did try to provide many helpful references to background material for any readers who found the material in the paper too dense.
>
> We appreciate that there may not be as many baselines in the paper as in some other papers on ML for chemistry. Ab-initio QMC calculations are extremely computationally expensive in comparison to many other methods - for instance, each benzene dimer experiment required more than a week of compute time on 64 A100 GPUs. This is because we are solving the Schroedinger equation directly, rather than learning to approximate existing data. If you look at other papers on deep neural networks for QMC, you will see we actually are investigating a relatively large number of systems. For instance, in Hermann, Schätzle and Noé (2020) (https://www.nature.com/articles/s41557-020-0544-y), they looked at only six systems, and that was accepted to Nature Chemistry. In a more recent work, Gerard et al. (2022) (https://openreview.net/forum?id=nX-gReQ0OT) looked at 17 different molecules and atoms, and that work was just accepted to NeurIPS with unanimously positive reviews. We investigated 19 different systems, and deliberately looked at heavier atoms than Gerard et al. (including zinc on top of the heavy atoms in their paper) and larger systems like naphthalene and CCl4. The computational cost for these larger, more ambitious, systems is substantial compared to the lighter atoms and smaller molecules in the earlier FermiNet (Pfau, Spencer, Matthews, Foulkes (2020)) and PauliNet (Hermann et al, ibid) papers. We felt it important to push neural network VMC Ansatzes in this direction and demonstrate the PsiFormer model can achieve substantial improvements on systems not previously studied. However, if you feel that is still not enough, we’d be happy to include a few more systems. Are there any systems in particular which you felt were excluded which you believe we should consider?
>
> We do also hope we can convince the reviewer that this work is much less incremental than other recent work in the area. In Gerard et al. (2022), the main innovation was to combine architectural details of the FermiNet and PauliNet, and found that this led to a 0.006 Ha improvement in the total energy of benzene. In the right panel of Fig. 3 in our paper, you can see that the PsiFormer makes a qualitative leap in performance relative to the results of Gerard et al. on benzene. On larger systems, such as those in Table 2, the leap over the FermiNet is even larger. In the comments to Reviewer x9Gw, we explain how significant the scale of improvement is relative to the standards of quantum chemistry. The architectural advance in our paper is also much greater than that in other recent work. For instance, Gerard et al. mainly combined parts of the FermiNet and PauliNet in a novel way. While self-attention is not novel in other fields of ML, it is a radical departure from existing architectures for ab-initio quantum chemistry, and (as highlighted by other reviewers) we believe that it is a significant contribution getting it to work for this application for the first time.

---

> ### Author Response · Authors · 2022-11-16
> **Awaiting the reviewer's response**
>
> There are now two days remaining in the period for discussion with authors. We hope we have provided as thorough and satisfactory a response to the reviewer's questions as was possible. We did hope to get some clarification from the reviewer on some of their points, and would still be open to addressing specific criticisms they have of the paper. If there are any further questions or clarifications that we could give, or any further changes to the paper that might raise the reviewer's opinion of it, please let us know.

---

> > ### Comment · Reviewer_58NZ · 2022-11-17
> > **The motivation cannot hold.**
> >
> > The motivation is  "but they lack an attention-like mechanism for gating interactions between electrons". I cannot agree with such an unprofessional reason. The ML community usually presents many expressive feature modeling tools each year. If this reason can hold, there will be more science works with such simple motivations. I encourage the authors to explore the connection between self-attention and structured electrons. This would be useful to explore their inherent relationships.

---

> > > ### Author Response · Authors · 2022-11-18
> > > **Thank you for the response**
> > >
> > > We thank the reviewer for their response, however we must object to their claim that the abstract of our paper is "unprofessional". We made two claims:
> > >
> > > 1. The FermiNet and PauliNet lack an attention-like gating mechanism
> > > 2. The PsiFormer is empirically a significant improvement over the FermiNet
> > >
> > > Both of these statements are factually correct (and the other reviewers all agree on the second point). We did not claim that the attention mechanism is *why* the PsiFormer works better than the FermiNet, we were only trying to highlight the difference between the PsiFormer and prior work. The value of the paper is entirely in the strength of the empirical results.

---

### Official Review · Reviewer_x9Gw · 2022-10-25

**Confidence:** 2
**Correctness:** 3
**Technical Novelty And Significance:** 2
**Empirical Novelty And Significance:** Not applicable
**Recommendation:** 6

**Clarity, Quality, Novelty And Reproducibility:**

Overall, the clarity of the experimental section can be much improved by explaining why the results in tables 1-3 are significant and how to interpret these results. The novelty of the work is moderate -- it uses self-attention and combine the pairwise features through Jastrow factors. It's also unclear why such design is the best. For example, why not use relational attention to incorporate pairwise features in the self-attention layer? Why do you include the pairwise features at the very end? Is this modeling choice guided by first principles?

**Strength And Weaknesses:**

The strength of this paper is that PsiFormer shows improvement over the FermiNet, especially on larger atoms with more electrons. The weakness, however, is mostly about the experiment section. There are many parts of the experiment that needs further clarification.
1. It's quite unclear whether the results are significant. The difference between FermiNet and PsiFormer is very close (within 0.01 for smaller atoms and 0.1 for larger atoms). Also, what's the ground truth for each electron system? Is it the result coming from VMC or DMC? If so, shouldn't you report the mean absolute error (MAE) between FermiNet/PsiFormer's prediction and VMC/DMC? Should a system get a lower MAE instead?
2. What's the level of accuracy (or precision) needed for an algorithm to be useful? Is FermiNet good enough already? (again, improving results by 0.01 seems very small but I can be wrong).
3. Why do you only report $\Delta E_{mono}$ and $\Delta E_{10A}$ for Benzene Dimer but not third-row atoms?

**Summary Of The Paper:**

This paper proposes a self-attention based architecture called PsiFormer for approximating many-electron Schrödinger equation. The architecture uses self-attention transformer to learn the embedding of electrons and the pairwise features are incorporated through Jastrow factor. The authors did a series of experiments on multiple molecular systems and compared PsiFormer with FermiNet, the previous state-of-the-art system for this task.

**Summary Of The Review:**

Overall, it is unclear to me how significant the results are and hard for me to draw a conclusion. I am not an expert in the field of quantum chemistry calculations so I am not confident in my judgment. I vote for weak rejection for now but open to suggestions during author feedback and discussion.

---

> ### Author Response · Authors · 2022-11-08
> **Response to Reviewer x9Gw (Part 1)**
>
> We thank the reviewer for their thoughtful comments, and we hope that we can convince the reviewer that the results are in fact significant. The reviewer had three major criticisms of the experiments section, which we will address in turn. Due to the length of our response, we are posting this as two separate comments. We address the reviewer's first concern in this comment, and the second and third concern in the second comment.
>
> The reviewer was uncertain about the exact significance of the results. While it is true that 0.1 to 0.01 Ha is a small amount relative to the total energy, this is in fact an enormous amount relative to the energy scales which chemists care about for calculations. We mention in the paper that the usual standard for a calculation to be “chemically accurate” is about 0.0016 Ha, or 1 kcal/mol - 0.1 Ha is roughly 60 times larger than that.  This was originally defined (or at least popularised) in John Pople’s Noble Prize lecture, and arises from the typical level of accuracy achievable in experiments. Futher, an accuracy of around 0.0016 Ha in relative differences  is required in order to correctly estimate reaction rates to within an order of magnitude. This is an enormously demanding level of accuracy, which is far more stringent than the level of accuracy expected in most ML applications, and presents a special challenge when applying deep learning to electronic structure problems.
>
> For a relevant example of what is considered “significant” in computational chemistry, we recommend the recent Science paper by Kirkpatrick et al. on improving density functionals (https://www.science.org/doi/full/10.1126/science.abj6511). Fig. 4 gives a comparison between benchmark methods. The improvement of the proposed DM21 functional is 0.5 kcal/mol, or 0.0008 Ha, on GMTKN55, and 44.6 kcal/mol, or 0.07 Ha, on BBB, and this was considered a significant enough advance to be accepted in Science. We should note that density functional theory is not directly comparable to the VMC methods considered in this paper (DFT is not an ab-initio wavefunction method), and a direct comparison of the PsiFormer on these benchmarks would not be practical, but the relevant energy scales should give a sense of what constitutes “significant” for the quantum chemistry community. Another example is the beryllium dimer, where the bonding was only recently resolved experimentally (Merritt et al., Science 2009 https://www.science.org/doi/10.1126/science.1174326), where the total bonding energy is ~0.004 Ha.
>
> The reviewer also asks about the ground truth for each of these systems. In fact, it is not possible to provide a true ground truth for the absolute energies for all but the smallest systems. Ground truth from experiment can sometimes be derived for relative energies by measuring the energy of formation for certain chemical reactions, but for larger systems even this can be difficult (as we note in the section on the benzene dimer, different experimental results give discrepancies on the order of 0.001 Ha for the binding energy). The advantage of VMC is that, because it gives an upper bound on the energy, we can guarantee that a lower energy is closer to the ground truth, but we cannot say how far it is from the ground truth. Other all-electron ab-initio methods, like coupled cluster, can give a decent estimate of the ground truth, and we use coupled cluster calculations as an approximate ground truth in Fig. 2. However, coupled cluster does not give an upper bound on the energy, so it may overshoot the ground truth in some cases.
>
> All FermiNet and PsiFormer calculations were done using VMC. For some of the systems studied, namely benzene and the benzene dimer, there are results in the literature for both VMC and DMC. We reported DMC results in this case to show just how significant the improvement is with the PsiFormer - in some cases, VMC with the PsiFormer is more accurate than DMC with the FermiNet, which is remarkable given that DMC is guaranteed to be more accurate than VMC with the same ansatz. However, none of these should be considered “ground truth” but only upper bounds of different quality. It also would not be meaningful to report MAE for a given system, because a given system should only have a single energy, so there is nothing to take the average over. MAE is a more meaningful metric when the number of systems being considered is so large that it is not practical to analyze all results individually (and when a ground truth value is available).
>
> (cont'd)

---

> > ### Author Response · Authors · 2022-11-08
> > **Response to Reviewer x9Gw (Part 2)**
> >
> > To the reviewer's second concern: as we mentioned in the above comment, and at the end of Sec 2.1 in the paper, the usual standard of accuracy expected by the quantum chemistry community is “chemical accuracy” - 1 kcal/mol or 1.6 mHa. However, this is typically the standard expected for relative accuracy between different systems - for instance, the reactants and products in a chemical reaction. We compare the ionization potentials of several third row atoms in the paper and find that, while the results for the potassium atom are well within chemical accuracy, the errors for transition metals (iron and zinc) are still a factor of 2-3 too large. This shows that, even though the PsiFormer is a large improvement in terms of absolute energy, there is still some way to go before it can be considered exact.
> >
> >
> > To the reviewer's third concern: $ΔE_{mono}$ and $ΔE_{10A}$ measure the difference in energy between isolated molecules and two molecules brought close together. In the case of third row atoms, we are not looking at the energies of two molecules (or atoms) brought close together, but only the energy of a single atom, with or without a single electron removed. While nuclear positions are fixed, electron positions are not, so if we tried to compute the ionization potential by placing a single electron far away, it would eventually just drift back to the nucleus. Therefore, the only way to estimate the ionization potential is to run a second calculation with the number of electrons reduced by one.
> >
> > We hope this addresses the individual points raised by the reviewer. As requested, we will rewrite the experiments section to make this clearer. The reviewer also asked about relational attention - a search through the literature found several different neural network architectures called “relational attention” and we would appreciate some clarification about which one the reviewer was asking about. We wanted to incorporate pairwise features of the electrons in the simplest possible way, in part because prior experience with the FermiNet showed that the two-electron stream is often the slowest part of the network. The Jastrow factor was in fact motivated by first principles - the Jastrow factor is chosen to exactly satisfy the Kato cusp conditions, which place strong constraints on the gradient of the wavefunction when two particles overlap. However, we are not claiming that this is the best possible way to incorporate pairwise information, only that it is a significant advance over the FermiNet and other VMC ansatzes. Alternative methods for incorporating pairwise input features are an excellent area of investigation for future research.

---

> > > ### Comment · Reviewer_x9Gw · 2022-11-17
> > > **Thank you for your response**
> > >
> > > Dear authors,
> > >
> > > Thank you for detailed feedback. I can see now that the results are significant, and I highly recommend authors to elaborate the results and their significance. However, I do feel that the architecture itself is not well motivated -- the design choice of Jastrow factor is interesting because it has connection to first principles, but that applies to FermiNet as well. That's why other reviewers have questions regarding the novelty of the architecture -- indeed, at the first glance, it's a just a simple application of self-attention...
> > >
> > > Overall, I do think this paper gives a good empirical contribution and I will increase my score to 6.

---

> > > > ### Author Response · Authors · 2022-11-18
> > > > **You're very welcome**
> > > >
> > > > Thank you for your considered reply and your openness to changing your mind about the paper. As highlighted by Reviewer GTMe, a main strength of our approach is that it really works, and is the first to do so. The use of self-attention is motivated by the electron correlation being a subtle and complicated interaction. The 1/r electron-electron interaction in the Hamiltonian does result in electron-electron interaction becoming weaker with distance, so a gating mechanism for this seems a reasonable approach. Self-attention has been extremely successful as a general way to introduce gating, and flexible wavefunction Ansatzes have proven to be much more accurate than those with fixed functional forms. Beyond this, it is difficult to say why self-attention is so effective here, though we look forward to exploring this question. Whilst space is very tight, we have added two sentences to the first paragraph in Section 3 discussing this.
> > > >
> > > > Similarly, space restricts us from extending the discussion on the importance of accuracy in quantum chemistry very much. We will rewrite the paper to better highlight what discussion we do already have. As a first step, we have added a sentence at the end of Section 2.1 signposting our results on the benzene dimer, where the binding energy is six orders of magnitude less than the total energy.

---

> ### Author Response · Authors · 2022-11-16
> **Awaiting the reviewer's response**
>
> There are now two days remaining in the period for discussion with authors. We hope we have provided a thorough and satisfactory answer to the reviewer's questions. If there are any further questions or clarifications that we could give, or any further changes to the paper that might raise the reviewer's opinion of it, please let us know.

---

### Author Response · Authors · 2022-11-08
**General Response**

We would like to thank all of the reviewers for the helpful comments, and hope that this can be the start of a productive discussion period. We have responded to individual reviewers’ concerns below, but also wanted to provide a general overview of the changes we have made to improve the paper and respond to feedback:

- Some reviewers suggested that the clarity of the experimental section could be improved, and hoped that we could go into more detail, for instance describing the MCMC algorithm used and relative run time of different architectures. Some of this was discussed in the appendix already, but we have included more details of the MCMC algorithm in the appendix, and we have updated the experimental section of the paper to provide a high level summary of the results in the appendix, along with pointers to the relevant sections of the appendix.
- Some reviewers requested more thorough experiments, for instance investigating more systems, or more detailed studies of the effect of different network configurations. While we have already included a relatively large number of systems in the paper compared to other papers on neural networks for QMC, we have expanded the experiments section with new results which were not available at the time of submission. For large molecules (benzene, toluene, naphthalene and CCl4) and the benzene dimer, we have included results both with and without LayerNorm, whereas previously only a single one was included. We believe this makes the relative contribution of different network configurations for different systems much clearer.

These were the changes we were able to make relatively quickly. For some of the more involved changes that the reviewers suggested, such as running more systems, or rewriting the experimental section in more depth to improve the clarity, we hope that during the discussion period we can reach an agreement about what changes would be acceptable to the reviewers. We look forward to answering any further questions you might have.

---

### Decision · Program_Chairs · 2023-01-20

**Decision:**

Accept: poster

**Justification For Why Not Higher Score:**

Although we decided to make an ACCEPT recommendation, one of the reviewers actually remain a conservative view on the paper.

**Justification For Why Not Lower Score:**

The paper concerns an important topic and has its merits in many ways.

**Metareview: Summary, Strengths And Weaknesses:**

This paper proposes a self-attention based architecture called PsiFormer for approximating many-electron Schrödinger equation. The architecture uses self-attention transformer to learn the embedding of electrons and the pairwise features are incorporated through Jastrow factor. The authors did a series of experiments on multiple molecular systems and compared PsiFormer with FermiNet, the previous state-of-the-art system for this task.

In general, this is a nice paper, with both algorithmic contribution and experimental advances. One reviewer raised his/her concern about the motivation and the explanability of the work. However, after several rounds of offline discussions among the reviewers, we believe these concerns are not that critical and it would be good to accept the paper to ICLR. We believe the audience will be interested in reading it.


**Note From Pc:**

if the above contains the word "oral" or "spotlight" please see: "oral" presentation means -> notable-top-5% and "spotlight" means -> notable-top-25%. As stated in our emails, we are disassociating presentation type from AC recommendations

**Summary Of Ac-Reviewer Meeting:**

GTMe:

In my opinion, it would be better to have more explanations, but it is not mandatory. Here are my reasons.
Usually there are two categories of explanations for why a particular NN structure works: (1) predictive explanations, (2) posterior explanations. Predictive explanations can tell you that your model will definitely work better (at least not worse) before you train your model, but posterior explanations only explain the reason once you have trained your model and found it better. As the structure of NN becomes more and more complicated, most papers only give posterior explanations because predictive explanations are almost impossible. In this paper, I think it will be too rigorous to require a predictive explanation, and useless to require a posterior explanation. So I think the paper is good for now.


58NZ:

I read the response. "The FermiNet and PauliNet lack an attention-like gating mechanism".- It cannot be a proper motivation. If so, once there is a new tool from the deep learning or ML community. There will be many papers like this. Even from an application perspective, they need more solid reasons to explain why they need such a new mechanism/tool/trick, and then explain why this mechanism/tool can help to achieve such performance.

I have the same feeling as the Reviewer x9Gw,---"The novelty of the work is moderate -- it uses self-attention and combines the pairwise features through Jastrow factors. It's also unclear why such design is the best.." And,  Reviewer RSBm also said that "The proposed idea itself is not very novel. To my knowledge, the novelty of this paper is limited to the self-attention mechanism and the Jastrow factor-based parameterization of electron-electron features. Both of them have already been considered in prior works, like the FermiNet."